# Can Compressed LLMs Truly Act? An Empirical Evaluation of Agentic Capabilities in LLM Compression

Peijie Dong [* 1]   Zhenheng Tang [* 2]   Xiang Liu [1]   Lujun Li [2]   Xiaowen Chu [1]   Bo Li [2]

## Abstract

Post-training compression reduces the computational and memory costs of large language models (LLMs), enabling resource-efficient deployment. However, existing compression benchmarks only focus on language modeling (e.g., perplexity) and natural language understanding tasks (e.g., GLUE accuracy), ignoring the agentic capabilities—*workflow*, *tool use/function call*, *long-context understanding* and *real-world application*. We introduce the **Agent Compression Benchmark (ACBench)**, the first comprehensive benchmark for evaluating how compression impacts LLMs' agentic abilities. ACBench spans (1) 12 tasks across 4 capabilities (e.g., WorfBench for workflow generation, Needle-in-Haystack for long-context retrieval), (2) quantization (GPTQ, AWQ) and pruning (Wanda, SparseGPT), and (3) 15 models, including small (Gemma-2B), standard (Qwen2.5 7B-32B), and distilled reasoning LLMs (DeepSeek-R1-Distill). Our experiments reveal compression tradeoffs: 4-bit quantization preserves workflow generation and tool use (1%–3% drop) but degrades real-world application accuracy by 10%–15%. We introduce *ERank*, *Top-k Ranking Correlation* and *Energy* to systematize analysis. ACBench provides actionable insights for optimizing LLM compression in agentic scenarios. The code can be found in `https://github.com/pprp/ACBench`.

## 1. Introduction

Large language models (LLMs) (Brown et al., 2020; Chiang et al., 2023; OpenAI, 2023; Team, 2024a) have demonstrated transformative capabilities in domains ranging from code synthesis (Roziere et al., 2023) and scientific research (Li et al., 2024d) to multi-agent collaboration (Wang et al., 2024a;b; Li et al., 2023a; Wu et al., 2024b). Despite these advances, their practical deployment remains hindered by prohibitive computational and memory costs (Samsi et al., 2023b; Stojkovic et al., 2024). Post-training compression techniques—notably pruning (Sun et al., 2024c; Dong et al., 2024b) and quantization (Park et al., 2024; Dong et al., 2024c)—address this challenge by reducing model size by up to $4\times$ while preserving performance on standard benchmarks like perplexity and arithmetic tasks (Zhang et al., 2024d).

Existing compression evaluations focus narrowly on single-turn language modeling (e.g., WikiText-2 perplexity) and natural language understanding (NLU) tasks (e.g., GLUE accuracy) (Gong et al., 2024; Yang et al., 2024a; Wang et al., 2025a; Lai et al., 2025). However, real-world agentic applications—such as robotic control (Li et al., 2023a) and financial analytics (Yang et al., 2023a)—demand capabilities that transcend these static benchmarks. Specifically, compressed LLMs must retain (1) multi-step planning (e.g., API call sequences in `WebShop`), (2) long-context coherence (e.g., 40k-token retrieval in `Needle-in-Haystack`), (3) adaptive reasoning across conversational turns in workflow, and (4) seamless tool integration (e.g., external API). Prior work (Li et al., 2024e) evaluates quantized models on NLU tasks but overlooks the interplay between compression and these agentic capabilities—a critical oversight given their centrality to deployment scenarios requiring multi-turn interactions.

We make a holistic evaluation of both quantized and pruned LLMs using Agent Compression Benchmark (ACBench) to reveal the status quo across three dimensions: (1) **Effects of compression on agent capabilities**: Whether compressed LLMs still perform well on other essential agent capabilities, such as planning, reasoning, tool use, and long-context understanding; (2) **Effect of compression on LLMs**: We employ three novel metrics to systematically analyze the gap between compressed and uncompressed LLMs - ERank, Top-K Ranking correlation, and energy. These metrics help reveal how compression influences model outputs and

---
[*]Equal contribution  [1]The Hong Kong University of Science and Technology (GuangZhou) [2]The Hong Kong University of Science and Technology. Correspondence to: Zhenheng Tang <zhtang.ml@ust.hk>, Xiaowen Chu <xwchu@hkust-gz.edu.cn>.

*Proceedings of the 42nd International Conference on Machine Learning*, Vancouver, Canada. PMLR 267, 2025. Copyright 2025 by the author(s).

Table 1: The benchmarks and model families for evaluation. "Size" denotes the test size, while "Env" denotes the number of environments.

| Section | Task & Ability | Benchmark | #Size/#Env | Model Family |
|---|---|---|---|---|
| Sec. 4 | Tool Use/T-Eval | Plan | 553 | InternLM-2.5-7B, Qwen-2.5-7B, Mistral-7B, DeepSeek-R1-Distill-Qwen, DeepSeek-R1-Distill-LLama |
| | | Reason | 6426 | |
| | | Retrieve | 64226 | |
| | | Understand | 6753 | |
| | | Instruct | 2660 | |
| | | Review | 487 | |
| Sec. 5 | Workflow Generation | Function Call Tasks | 1803 | Qwen-2.5(1.5B-32B), InternLM-2.5-7B, Llama-3.1-8B,Mistral-7B, DeepSeek-R1-Distill-Qwen(1.5B, 7B), DeepSeek-R1-Distill-Llama-8B |
| | | Embodied Tasks | 4048 | |
| | | Problem-Solving Tasks | 4257 | |
| | | Open-Grounded Tasks | 2281 | |
| Sec. 6 | Code | Lcc, RB-P | 1000 | InternLM-2.5-7B, Qwen-2.5(1.5B, 3B, 7B), Megrez-3B, MiniCPM-4B, Gemma-2B, Phi-3.5, DeepSeek-R1-Llama-8B, DeepSeek-R1-Distilled(1.5B, 7B) |
| | Single-Doc QA | NrtvQA, Qasper, MF-en | 750 | |
| | Multi-Doc QA | HotpotQA, 2WikiMQA, Musique, Dureader | 800 | |
| | Summarization | QMSum, MultiNews, VCSum | 600 | |
| | Few-shot | TREC, TriviaQA, SAMSum, LSHT | 800 | |
| | Synthetic | PRE, Pcount | 600 | |
| | LongGenBench | GSM8K | 8000 | |
| | | MMLU | 15908 | |
| | Information Extraction | Needle-in-the-Haystack | 40K | |
| Sec. 7 | Embodied AI | ScienceWorld | 90 | Qwen-2.5(1.5B-7B), InternLM-2.5-7B, DeepSeek-R1-Distilled-Qwen-7B, DeepSeek-R1-Distilled-Qwen-1.5B, DeepSeek-R1-Distilled-Llama-8B |
| | Game | Jericho | 20 | |
| | | PDDL | 60 | |
| | Tool Use | Tool-Query | 60 | |
| | | Tool-Operation | 40 | |

internal representations, providing insights into which compression configurations best preserve agent capabilities. (3) **Impact of different compression approaches**: The relative effectiveness of quantization versus pruning methods, and their potential complementarity, remains unexplored in the context of agent tasks. A comprehensive comparison is needed to guide optimal compression strategy selection.

Specifically, we evaluate three categories of LLMs: (1) Small language models (<7B parameters) like Phi-3.5 (Abdin et al., 2024), MiniCPM (Hu et al., 2024), and Megrez (Infinigence, 2024); (2) Standard models (7B-32B parameters) including Qwen2.5, InternLM2.5, and Mistral-7B; and (3) Distilled Reasoning models like DeepSeek-R1-Distilled series. For agent tasks, we systematically evaluate four core agentic capabilities (Action Execution, Workflow Generation, Long-Context Understanding, and Real-world Application) that enable complex workflow execution while maintaining coherent behavior across extended interactions. To investigate the effects of compression configuration, we evaluate two mainstream compression methods: quantization (including GPTQ (Frantar et al., 2022) and AWQ (Lin et al., 2023)) and pruning (including SparseGPT (Frantar & Alistarh, 2023a) and Wanda (Sun et al., 2024b)).

To facilitate systematic analysis of compression impacts, we introduce three novel analytical tools: (1) Efficient Rank(Roy & Vetterli, 2007; Wei et al., 2024) leverages

rank-based techniques to capture the distribution of model logits more efficiently. This tool helps identify the structural changes in the model's decision-making process induced by compression. (2) Top-K Ranking Correlation can measure the ranking consistency between the top-k logits of compressed and uncompressed LLMs, providing insights into how well the compressed model preserves the original model's confidence in its predictions. (2) Energy-based Analysis, inspired by Out-of-Distribution (OOD) detection techniques (Lee et al., 2018; Liu et al., 2020), evaluates the distributional shifts in logit energies between the compressed and uncompressed models.

## 2. Preliminaries

In this paper, we mainly focus on quantization and weight pruning as presented in Figure 1(b), with additional experiments on distilled DeepSeek-R1 series. Additionally, we design three metrics to measure the gap between compressed and uncompressed LLMs.

**Quantization** reduces the memory and computational demands of neural networks by mapping full-precision values (e.g., 16-bit floats) to lower-bit integer representations. The affine transformation

$$\mathbf{X}_{\text{INT}} = \text{round}\left(\frac{\mathbf{X}_{\text{FP16}} - Z}{S}\right) \tag{1}$$

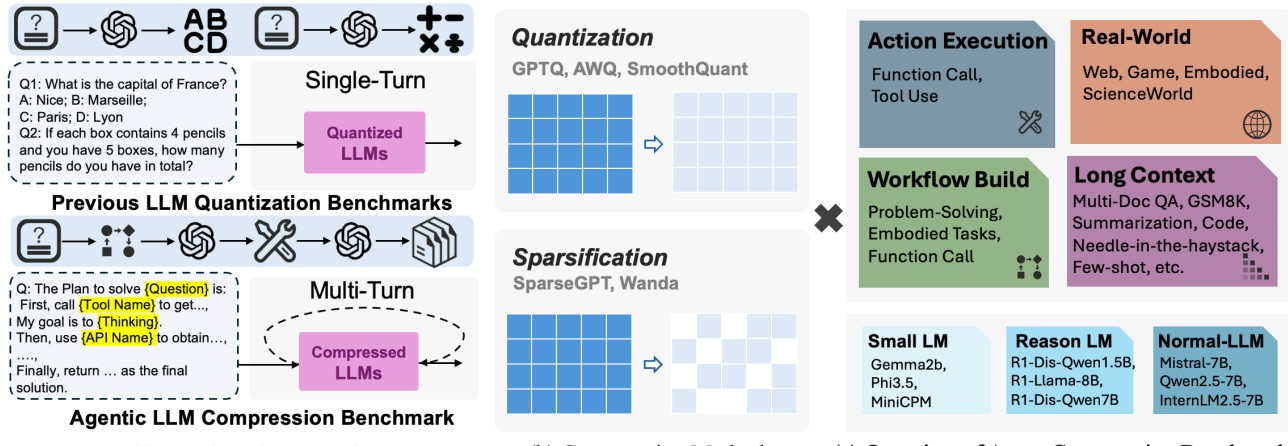

(a) Benchmark Comparison  (b) Compression Methods  (c) Overview of Agent Compression Benchmark

Figure 1: Overview of Agent Compression Benchmarks and Methods for Large Language Models (LLMs). (a) Benchmark Comparison: Illustrates the transition from single-turn quantized LLMs to multi-turn compressed LLMs in agentic scenarios. (b) Compression Methods: Summarizes the techniques used for quantization (e.g., GPTQ, AWQ, SmoothQuant) and sparsification (e.g., SparseGPT, Wanda). (c) Overview of Agent Compression Benchmark: Provides a comprehensive view of the capabilities and components involved in agentic LLM compression, including action execution, workflow build, real-world applications, and long-context processing.

$$S = \frac{\max(\mathbf{X}_{\text{FP16}}) - \min(\mathbf{X}_{\text{FP16}})}{2^{N-1} - 1} \quad (2)$$

preserves the dynamic range of the original tensor $\mathbf{X}_{\text{FP16}}$, where $N$ specifies the target integer bit-width (e.g., 8 bits), $S$ is the scaling factor, and $Z$ aligns the integer zero-point with the floating-point range. In this paper, we mainly focus on GPTQ (Frantar et al., 2022) and AWQ (Lin et al., 2023).

**Weight Pruning** removes redundant parameters to create sparse weight matrices, reducing model size and inference latency. Unstructured sparsity is achieved via element-wise masking:

$$\hat{\mathbf{W}} = \mathbf{W} \odot \mathbf{M}, \quad \mathbf{M}_{ij} = \begin{cases} 1 & \text{if } |\mathbf{W}_{ij}| > \tau \\ 0 & \text{otherwise} \end{cases} \quad (3)$$

where $\tau$ governs the sparsity level. For hardware efficiency, structured pruning eliminates entire channels using the $L_1$-norm $\sum_i |\mathbf{W}_{c,i}|$ to identify less salient features, trading finer granularity for better computational alignment. In this paper, we mainly focus on SparseGPT (Frantar & Alistarh, 2023a) and Wanda (Sun et al., 2024b) with unstructured and 2:4 semi-structured settings.

### 2.1. Agentic Taxonomy

Since there are already several benchmarks (Li et al., 2024e; Yang et al., 2024a; Gong et al., 2024; Wang et al., 2024a;b) that focus on the capabilities of LLMs like in-context learning, commonsense multi-step reasoning, mathematical multi-step reasoning, instruction following, and self-calibration. However, the agentic capabilities of open-

sourced LLMs have not been well-studied. To fill the gap, we focus on the distinct characteristics of agentic capabilities in this paper. To study the influences of compression on the agentic capabilities of LLMs, we mainly focus on the capabilities that can be influenced by compression. We classified the agentic capabilities into four main ones as shown in Figure 1(c) and Table 1:

(1) **Action Execution** includes function call and tool use (Chen et al., 2023c). Function call denotes that the agent can employ the user predefined functions to enhance the capabilities, while tool use denotes that the agent can use the external tools to enhance the capabilities. We focus on the six key capabilities: plan, reason, retrieve, understand, instruct, and review.

(2) **Workflow Generation** enables agents to break down complex tasks into executable sequences of steps (Wang et al., 2024a;b). This includes both single-task workflows where a direct solution path exists, and multi-step workflows requiring intermediate planning and coordination across multiple tools or APIs. We mainly focus on four key tasks: function call, embodied, problem-solving, and open-grounded tasks.

(3) **Long-Context Understanding**: Due to the nature of multi-step workflows, the context length is obviously large compared with basic NLP tasks. In the near future, as the context length is further increased, the long-context understanding will become a key capability for agentic applications. We mainly focus on general long-context tasks like single- and multi-doc QA, summarization, and

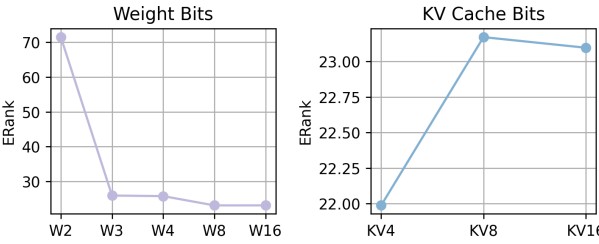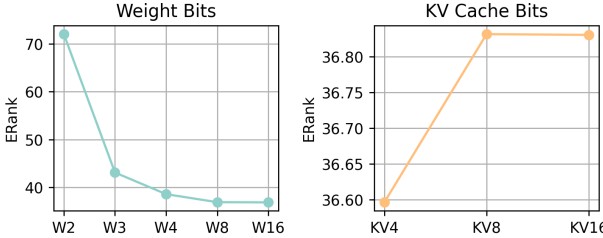

Figure 2: ERank analysis difference analysis for quantized LLaMA-2-7B (left) and Mistral-7B (right) models

code, etc., and the more challenging ones like LongGen-Bench (Liu et al., 2024a) (GSM8K and MMLU), Needle-in-the-Haystack.

(4) **Real-world Application** encompasses the agent's ability to operate in practical deployment scenarios (Wang et al., 2025a) like e-commerce, robotics control, and scientific experimentation. This requires coordinating multiple capabilities including tool use, planning, and environmental interaction. We evaluate performance on benchmarks like EmbodiedAI (ScienceWorld), Game (Jericho, PDDL), and practical tool use including Tool-Query and Tool-Operation. These tasks can faithfully give us feedback on how agents can process the real-world applications.

## 2.2. Benchmarks and Models

Our experimental evaluation focuses on compressed language models, encompassing both quantized and pruned variants. We assess their performance across four distinct categories of agent tasks: Action Execution, Workflow Generation, Long-Context Understanding, and Real-world Application. In our investigation, we specifically prioritize compression algorithms that are compatible with contemporary inference frameworks such as vLLM (Kwon et al., 2023), thereby ensuring practical deployability and efficient model serving in real-world applications.

**LLMs.** To ensure a comprehensive evaluation, we assess the performance of state-of-the-art language models, prioritizing their latest versions to maintain relevance. Our analysis focuses on models with demonstrated agentic capabilities, as earlier models generally lack these essential features. We categorize the evaluated models into three groups: (1) Medium-scale models (7B parameters): InternLM-2.5-7B (Cai et al., 2024), Qwen2.5-7B (Yang et al., 2024b), and Mistral-7B (Jiang et al., 2023a); (2) Knowledge-distilled models: DeepSeek-R1-Distill-Qwen-1.5B, DeepSeek-R1-Distill-Qwen-7B, and DeepSeek-R1-Distill-Llama-8B (Team, 2024a); (3) Efficient small-scale models: MiniCPM3-4B (Hu et al., 2024), Qwen-2.5-1.5B, Qwen-2.5-3B (Yang et al., 2024b), Gemma-2-2b-it (Team, 2024b), Phi-3.5-mini-3.8B (Abdin et al., 2024), and Megrez-

3B (Infinigence, 2024).

**Datasets.** Instead of using standard datasets like previous benchmarks such as llmc (Gong et al., 2024), we uniquely focus on agentic tasks. We categorize the agentic tasks into three types: (1) Action Execution: T-Eval (Chen et al., 2023c), ToolBench (Qin et al., 2023a), and ToolAlpaca (Tang et al., 2023b); (2) Workflow Generation: Worf-Bench (Qiao et al., 2024); (3) Long-Context Understanding: LongBench (Bai et al., 2024), LongGenBench (Liu et al., 2024a), and Needle-in-the-haystack (Kamradt, 2023); (4) Real-world applications: AgentBoard (Ma et al., 2024). For calibration data, to ensure a fair comparison, we follow the setting in previous works (Gong et al., 2024) and employ the same subset of the Pile (Gao et al., 2020) validation set. We set the calibration data size to 128 and the sequence length to 512.

**Compression Methods.** We focus on post-training compression methods, including quantization and pruning. We use the following quantization methods: (1) GPTQ (Frantar et al., 2022); (2) AWQ (Lin et al., 2023); (3) SmoothQuant (Xiao et al., 2023). We employ unstructured, structured, and semi-structured pruning methods. SparseGPT (Frantar & Alistarh, 2023a) and Wanda (Sun et al., 2024c) are used for unstructured and semi-structured pruning. To ensure fairness and reproducibility, we set the temperature to 0.

## 2.3. Statistical Analysis

To investigate the influences of compression on the LLMs and how the compression affects the LLMs, we employ three statistical analysis metrics:

(1) **Efficient Rank (ERank):** The effective rank of a non-zero matrix $\mathbf{A} \in \mathbb{R}^{d \times N}$ is defined as

$$\text{eRank}(\mathbf{A}) = \exp\left(-\sum_{i=1}^{Q} \frac{\sigma_i}{\sum_{j=1}^{Q} \sigma_j} \log\left(\frac{\sigma_i}{\sum_{j=1}^{Q} \sigma_j}\right)\right), \tag{4}$$

where $Q = \min\{N, d\}$, and $\sigma_1, \sigma_2, \ldots, \sigma_Q$ denote the singular values of the matrix $\mathbf{A}$. This measure quantifies the distribution of the singular values and provides a notion of

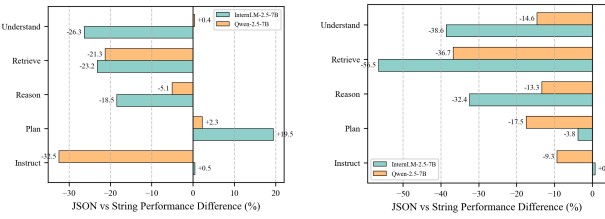

Figure 3: Comparison of format performance differences between (left) quantized and (right) sparse model architectures

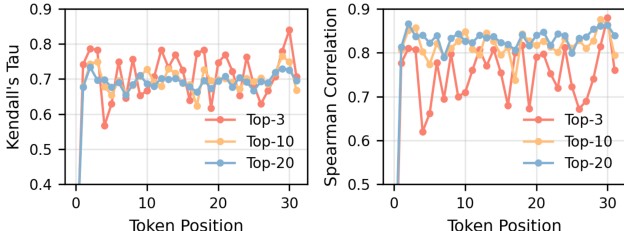

Figure 4: Top-k Ranking Consistency Analysis for quantized Phi-3.5.

the matrix's effective dimensionality.

(2) **Top-K Ranking Consistency:** For a given input, let $\mathcal{T}_k^{(o)}$ and $\mathcal{T}_k^{(c)}$ be the sets of top-k tokens according to $\mathbf{z}^{(o)}$ and $\mathbf{z}^{(c)}$ respectively. We measure ranking consistency using the Jaccard similarity:

$$J_k = \frac{|\mathcal{T}_k^{(o)} \cap \mathcal{T}_k^{(c)}|}{|\mathcal{T}_k^{(o)} \cup \mathcal{T}_k^{(c)}|} \tag{5}$$

(3) **Energy-based Analysis:** Many OOD detection methods (Lee et al., 2018; Liu et al., 2020) utilize the energy-based model to calculate the energy score as the confidence of the model on the input data $\mathbf{x}$. Let $f(\mathbf{x}) : \mathbb{R}^D \to \mathbb{R}^K$ be a discriminative neural classifier that maps an input $\mathbf{x} \in \mathbb{R}^D$ to $K$ logits. For both original model $f^{(o)}$ and compressed model $f^{(c)}$, the categorical distribution is derived using the softmax function:

$$p(y|\mathbf{x}) = \frac{e^{f_y(\mathbf{x})/T}}{\sum_{i=1}^K e^{f_i(\mathbf{x})/T}} \tag{6}$$

where $f_y(\mathbf{x})$ indicates the $y$-th logit corresponding to class label $y$, and $T$ is the temperature parameter. The energy function $E(\mathbf{x}; f)$ for a given input $\mathbf{x}$ is defined as:

$$E(\mathbf{x}; f) = -T \cdot \log \sum_{i=1}^K e^{f_i(\mathbf{x})/T} \tag{7}$$

We compare the energy distributions between original and compressed models using $\Delta_E = |E(\mathbf{x}; f^{(o)}) - E(\mathbf{x}; f^{(c)})|$, where lower $\Delta_E$ indicates better preservation of model confidence patterns.

## 3. Statistical Analysis of Compression Effects

**Efficient Rank Analysis.** We visualize the ERank (Wei et al., 2024) of LLaMA-2-7B and Mistral-7B in Figure 2. We separately evaluate the ERank under varying weight quantization levels (W2, W3, W4, W8, W16) and KV Cache precisions (KV4, KV8, KV16). Our findings indicate that both models exhibit similar trends in their effective rank

across these configurations. Specifically, the ERank decreases as the quantization precision reduces, suggesting a loss of information and structural complexity in the weight matrices. This behavior highlights the impact of quantization on the intrinsic dimensionality of the models' representations, which may contribute to the performance trade-offs observed in downstream tasks. As shown in Table 2, this relationship holds across model scales - from 125M to 6.7B parameters, where we observe ERank values ranging from 13.898 to 17.877 for 4-bit quantized models. Notably, the ERank values correlate positively with model accuracy, with larger models generally maintaining higher ERank values post-compression. For instance, the 6.7B model achieves the highest ERank of 17.877 and correspondingly strong accuracy of 0.360, while the 2.7B model shows a lower ERank of 13.898 despite having comparable accuracy. This suggests that larger models may be more robust to compression, maintaining their structural complexity better. The Diff-ERank metric, measuring the change in effective rank after compression, also shows a consistent trend, increasing with model size from 1.410 to 2.280, indicating that larger models undergo more significant structural changes during compression while still preserving performance.

**Top-K Ranking Consistency Analysis.** We quantitatively assess the discrepancy between compressed and uncompressed LLMs through top-k ranking consistency metrics. As depicted in Figure 4, we evaluate the ranking consistency using both Kendall's Tau and Spearman Correlation coefficients. Our analysis reveals a notable pattern: as $k$ decreases from 10 to 3, the ranking consistency exhibits increasing instability and degradation. This finding is particularly significant for LLM text generation, where the top-3 tokens are crucial as they represent the most probable next-token predictions. This degradation in ranking consistency for the most probable tokens provides a potential explanation for the observed performance deterioration in downstream tasks when using compressed LLMs. Additionally, Figure 5 demonstrates the strong Spearman correlation between perplexity and Top-k ranking correlation metrics across different model sizes and quantization levels, validating the metrics' ability to capture meaningful performance characteristics.

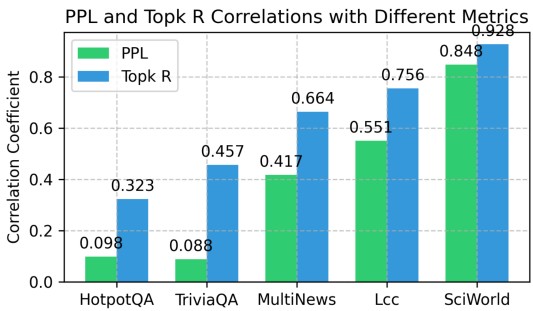

Figure 5: Spearman correlation analysis between perplexity and Top-k ranking correlation metrics across model sizes and quantization levels.

| OPT | 125M | 1.3B | 2.7B | 6.7B |
|---|---|---|---|---|
| ACC | 0.276 | 0.332 | 0.370 | 0.360 |
| $\Delta$ Loss | 5.734 | 6.138 | 6.204 | 6.258 |
| Diff-ERank | 1.410 | 2.140 | 2.338 | 2.280 |
| ERank (4bit) | 15.462 | 15.589 | 13.898 | 17.877 |
| Energy | 2.738 | 2.746 | 2.631 | 2.883 |

Table 2: Comparison of metrics across model sizes demonstrating correlation between ERank, Energy and model performance.

**Energy-based Analysis.** We visualize the distribution of energy scores of compressed and uncompressed LLMs for a series of tokens. As shown in Appendix C.7, we observed that the quantized LLM has a distinct energy distribution compared with uncompressed ones in the initial stage, while they begin to merge in the subsequent stage. The higher negative energy means high confidence in the prediction. The results show that in the initial decoding stage of the compressed LLM, the confidence distribution is polarized, with some over-confidence on certain tokens and some under-confidence on others. During the late decoding stage, both compressed and uncompressed LLMs regard tokens as low confidence. As shown in Table 2, we observe consistent energy scores across different model sizes, indicating that this behavior is inherent to the compression process rather than model scale dependent.

## 4. Evaluation on Action Execution

### 4.1. Experimental Setups

We evaluate the impact of quantization and pruning on LLMs' agent action execution capabilities, focusing on function calling and tool use. We utilize T-Eval (Chen et al., 2023c), which assesses six core competencies: planning, reasoning, retrieval, understanding, instruction following, and reviewing. Additional results can be found in Table 8 in Appendix C.2.

### 4.2. Effects of Compression

**Quantization Under Same Bit Budget.** Figure 7 presents a comparative analysis of model performance under equivalent bit budgets. For instance, a 7B model in FP16 and a 14B model in FP8/INT8 occupy similar memory footprints, enabling us to evaluate the effectiveness of quantization versus training larger models from scratch at FP16 precision. While this comparison cannot perfectly control for all variables such as training data and quantization methods, it reveals important trends. For smaller model sizes (1.5B-3B parameters), base models trained from scratch with BF16 precision generally demonstrate superior performance. However, as model size increases to around 7B parameters, quantized versions of larger models (e.g., AWQ-quantized 32B models) begin to show significant advantages. This suggests that quantization becomes increasingly valuable for larger model architectures, highlighting its crucial role in making larger, more capable models practically deployable while maintaining strong performance characteristics.

**Model Compression Significantly Impacts Structured Output Generation.** Our experimental results in Table 8 demonstrate significant performance variations between JSON and string format outputs under different compression techniques. The analysis reveals that model performance consistently degrades more severely when generating JSON-structured outputs compared to string formats. As evidenced in Figure 3, this performance disparity is particularly pronounced in both InternLM2.5-7B and Qwen2.5-7B architectures.

**Quantization Preserves Tool Use Capabilities Better Than Sparsification Methods.** The empirical evidence presented in Table 8 indicates that quantization approaches, specifically GPTQ and AWQ, maintain model performance more effectively than sparsification techniques such as SparseGPT and Wanda. However, it is noteworthy that Wanda (Sun et al., 2024c) achieves comparable performance to quantization methods when implemented with unstructured pruning parameters.

**Tool Use Capabilities Vary Significantly Across Model Architectures.** From Table 8, the experimental results reveal substantial performance differentials among model architectures, with InternLM2.5-7B and Qwen2.5-7B demonstrating superior tool use capabilities compared to Mistral-7B. This observation suggests that architectural advancements in newer models contribute significantly to enhanced tool use proficiency.

**Distillation from a reasoning model lead to performance degradation in agent scenarios.** Despite the theoretical advantages of incorporating explicit reasoning processes, our experiments with the DeepSeek-R1-Distilled variant of Qwen2.5-7B reveal a degradation in performance. As shown

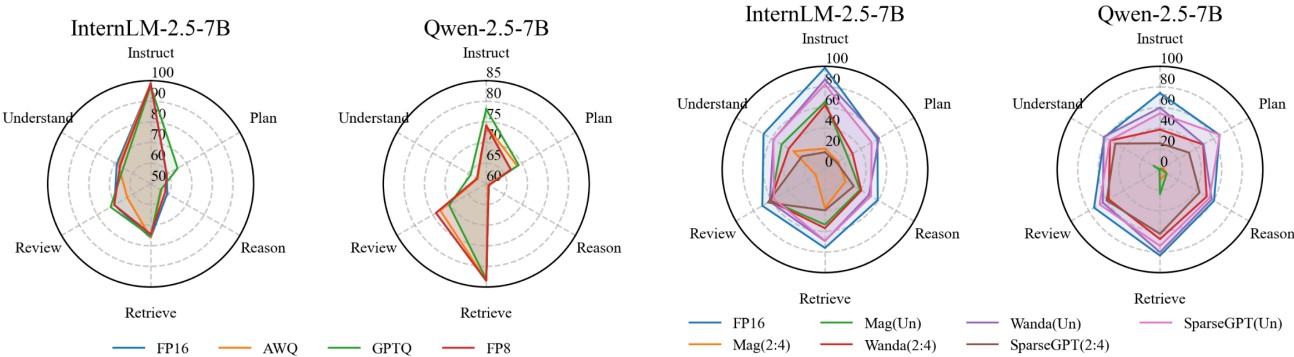

Figure 6: Performance Evaluation of InternLM-2.5-7B and Qwen-2.5-7B Models Across Different Sparsification (left) and Quantization (right) Techniques for Tool Use

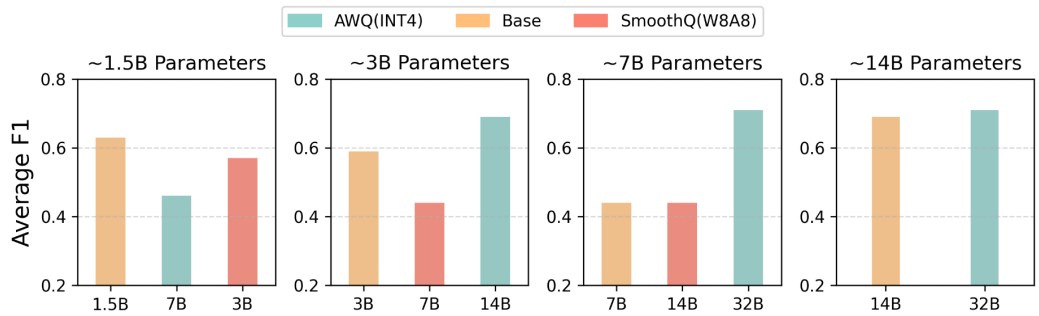

Figure 7: Comparison of average F1 scores across different model sizes and configurations. The models are evaluated based on their performance with approximately 1.5B, 3B, 7B, and 14B parameters. The configurations include AWQ(INT4), FP16, GPTQ(INT4), GPTQ(INT8), and SmoothQ(W8A8). Each bar represents the average F1 score achieved by the respective model configuration at different parameter sizes.

in Table 8, we observe a substantial decline in average accuracy from 70% to 43.6%, contradicting conventional expectations regarding the benefits of knowledge distillation.

**Compression Strategy Recommendations.** Based on our empirical analysis, quantization emerges as the most effective compression approach for maintaining tool use capabilities across diverse scenarios. When available, GPTQ with 4-bit precision or FP8 representation demonstrates superior performance characteristics. For applications requiring sparsification, our findings suggest that Wanda with 2:4 sparsity patterns provides an optimal trade-off between model performance and compression efficiency.

## 5. Evaluation on Workflow Generation

### 5.1. Experimental Setups

Our experimental evaluation is conducted using Worf-Bench (Qiao et al., 2024), a comprehensive benchmark that employs a graph-based workflow representation. This framework enables us to systematically assess both workflow gen-

eration capabilities and execution performance in multi-turn conversational settings. The evaluation encompasses four distinct categories of tasks: function calling, embodied interaction, problem-solving, and open-grounded scenarios. For compression techniques, we evaluate three quantization methods - AWQ (Lin et al., 2023), GPTQ (Frantar et al., 2022), and SmoothQuant (Xiao et al., 2023), alongside three pruning approaches - Magnitude pruning, SparseGPT (Frantar & Alistarh, 2023a), and Wanda (Sun et al., 2024b). Detailed experimental results are presented in Table 10 in Appendix C.4.

### 5.2. Effects of Compression

**Model Compression Shows Minimal Impact on Workflow Generation Capabilities.** Our experimental results, as presented in Table 10, demonstrate that most compression methods maintain model performance within a 5% degradation margin, with only magnitude-based pruning showing significant performance deterioration. This resilience suggests that workflow generation tasks primarily rely on high-level planning capabilities that remain largely intact

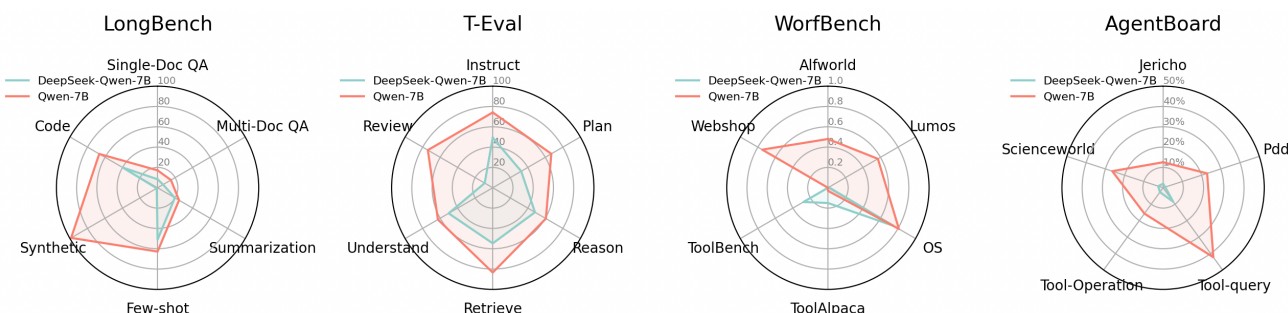

Figure 8: Performance Comparison between DeepSeek-R1-Qwen2.5-7B and Qwen2.5-7B across four evaluation benchmarks. The distilled version performs worse in most cases.

under compression. Among the evaluated architectures, Mistral-7B and Qwen2.5-7B demonstrate notably superior performance compared to InternLM2.5-7B, indicating that architectural differences may be more influential than compression effects.

**Robust Performance Maintained for OS and Webshop Tasks Under Compression.** Our analysis reveals that models maintain approximately 80% performance across all compression methods for operating system and e-commerce tasks. This robustness likely stems from the structured nature of these domains and the relatively constrained action space. Based on these findings and considering the compute efficiency trade-offs, we recommend GPTQ and AWQ as optimal compression choices for these applications.

**Model Size Influences Compression Sensitivity in Specialized Tasks.** In Alfworld and Lumos tasks, which require fine-grained language understanding and complex reasoning, we observe a clear correlation between model size and compression resilience. Smaller architectures, such as Qwen2.5-3B, experience substantial performance degradation of up to 50% under GPTQ and AWQ compression. In contrast, larger models like Qwen2.5-32B maintain their performance under quantization, occasionally showing slight improvements. This suggests that larger models possess redundant capacity that helps preserve critical capabilities under compression.

**Distilled Models Shows inferior Performance than undistilled Version.** Our evaluation of DeepSeek-R1 distilled models reveals significant performance regression compared to their undistilled versions. For example, DeepSeek-R1-Distilled-Qwen2.5-7B achieves only a 20% average F1 score, substantially lower than the 44% achieved by the undistilled Qwen2.5-7B. This counter-intuitive finding suggests that current distillation techniques may not effectively preserve complex reasoning capabilities. Interestingly, the smaller DeepSeek-R1-Distilled-Qwen2.5-1.5B outperforms its larger 7B and 8B counterparts, indicating that model size alone does not determine distillation effectiveness. Fur-

thermore, we observe that the uncompressed Megrez-3B demonstrates insufficient capability for workflow generation tasks, highlighting the importance of architectural design beyond parameter count.

## 6. Evaluation on Long-Context Understanding

### 6.1. Experimental Setups

We evaluate long-context capabilities across three benchmarks: (1) **LongBench** (Bai et al., 2024), a multi-task benchmark spanning Single/Multi-Doc QA, Code Comprehension, Synthetic Reasoning, Summarization, and Few-shot Learning, with analyses of quantization (Table 3), sparsification (Table 4), and small/distilled models (Table 5); (2) **LongGenBench** (Liu et al., 2024a), designed for extreme-length generation with multi-shot in-context examples, tested via quantized/pruned models (Table 6) and lightweight architectures (Table 7); (3) **Needle-in-the-Haystack** (Kamradt, 2023), a retrieval-focused task probing fine-grained contextual understanding, visualized across varying context lengths in Appendix C.5.

### 6.2. Effects of Compression

**LongBench Analysis:** Our experimental results demonstrate that quantization and sparsification exhibit minimal impact on few-shot learning, synthetic task performance, and code completion capabilities for models exceeding 7B parameters. As evidenced by Table 3 and 4, most compression methods induce negligible performance degradation, with the exception of magnitude-based sparsification. However, smaller architectures (Qwen2.5-1.5B, Qwen2.5-3B, MiniCPM-4B, and Gemma-2B) exhibit significantly reduced baseline capabilities, often failing to complete fundamental tasks even prior to compression. Notably, the Merge-3B model achieves state-of-the-art performance among sub-7B models. The DeepSeek-R1-Distilled series are particular sensitivity to compression, with substantial performance drop in single-document QA, multi-document QA, and syn-

thetic tasks compared to uncompressed ones.

**LongGenBench Findings:** In a more challenging benchmark LongGenBench, AWQ consistently outperforms other compression methodologies. While Wanda and SparseGPT achieve competitive performance with AWQ on Qwen2.5-7B (Table 7), this parity does not extend to InternLM2.5-7B, where only AWQ maintains baseline-equivalent performance. Intriguingly, compressed Qwen2.5-3B models demonstrate MMLU scores comparable to 7B counterparts, yet suffer catastrophic performance collapse in GSM8K reasoning (61% → 11%). Smaller models (<3B) exhibit near-zero accuracy on both MMLU and GSM8K. A notable discrepancy emerges between Qwen2.5-7B (61% GSM8K, 64% MMLU) and its DeepSeek-R1-Distilled variant (near-zero scores), suggesting that reasoning patterns may significantly impact long-context comprehension capabilities.

**Needle-in-the-Haystack Evaluation:** Both quantization and sparsification adversely affect long-context information retrieval, as detailed in Appendix C.5. Magnitude-based pruning induces particularly severe performance degradation. Our 40K-context experiments reveal consistent performance boundaries at 32K tokens across all compressed models, suggesting architectural limitations inherent to LLM designs and training paradigms.

## 7. Evaluation on Real-World Applications

### 7.1. Experimental Setup

Our empirical evaluation of compressed LLMs encompasses real-world agent tasks utilizing the **AgentBoard** framework (Ma et al., 2024). The evaluation spans three primary domains: (1) **Embodied AI**, with a focus on the *ScienceWorld* environment for physical interaction simulation; (2) **Game Interaction**, incorporating both *Jericho* text-based adventures and *PDDL* planning scenarios; and (3) **Tool Use**, which examines both tool querying and operational capabilities. For more details, please refer to the experimental results in Table 9 of Appendix C.3.

### 7.2. Effects of Compression

**Compressed LLMs face significant challenges in handling most real-world scenarios.** Through comprehensive evaluation detailed in Table 9, we assessed various quantization and pruning techniques applied to 7B LLMs. The results demonstrate that these compressed models generally exhibit substantial degradation in real-world task performance. Among the compression methods evaluated, only AWQ and Wanda maintained acceptable performance levels, while alternative approaches showed marked deterioration in capability across multiple task domains.

**DeepSeek-R1-Distilled model series exhibited minimal**

**progress and success rates in practical applications.** As illustrated in Table 9, the distilled models experienced a significant degradation in functional performance across practical tasks. For instance, the progress rate on the Pddl benchmark for the **DeepSeek-R1-Distilled-Qwen2.5-7B** model dropped markedly from 33% to 1%. In contrast, smaller models such as **Qwen2.5-3B**, when enhanced with AWQ, achieved a 23% progress rate. This stark performance gap can be attributed to two key factors. First, the DeepSeek-R1 teacher model used for distillation lacked robust agentic capabilities in its training, particularly for tool use and practical task execution. As a result, the distillation process could not effectively transfer these crucial agentic skills to the student model. Second, the limited model capacity creates an inherent trade-off during distillation - when the process prioritizes core reasoning skills like mathematical problem-solving, it necessarily deprioritizes the preservation of agentic capabilities like function calling and tool manipulation. As presented in Figure 8, we compare the undistilled **Qwen2.5-7B** with its distilled counterpart across LongBench, T-Eval, WorfBench, and AgentBoard, revealing consistent degradation in practical task performance despite improvements in traditional reasoning tasks. These findings highlight the need to develop distillation techniques that can better balance the preservation of both abstract reasoning and practical agentic capabilities.

## 8. Conclusion

In this paper, we introduce ACBench, the first benchmark for evaluating LLM compression's impact on agentic capabilities, including action execution, workflow generation, long context understanding and some real-world applications. By introducing ERank, Top-k Correlation, and Energy metrics, we systematize compression analysis for agent tasks. ACBench provides a practical tool for efficient LLM deployment, and inspires future LLM compression to focus more on real-world LLM based agentic applications.

## 9. Limitations

Our study is limited to examining post-training quantization and sparsification for agentic capabilities, and does not involve quantization-aware training (QAT) approaches. We restrict our analysis to compression methods compatible with vLLM (Kwon et al., 2023), excluding promising techniques like QuaRot (Ashkboos et al., 2024) due to their computational overhead. Additionally, we employ default configurations throughout our experiments without exploring variations in parameters such as group size. Consequently, our findings and recommendations should be interpreted within the context of these specific experimental conditions and may not necessarily extend to other scenarios or tasks.

## Acknowledgments

This work was partially supported by National Natural Science Foundation of China under Grant No. 62272122, the Guangzhou Municipal Joint Funding Project with Universities and Enterprises under Grant No. 2024A03J0616, Guangzhou Municipality Big Data Intelligence Key Lab (2023A03J0012), Hong Kong CRF grants under Grant No. C7004-22G and C6015-23G, a RGC RIF grant under contract R6021-20, RGC TRS grant under contract T43-513/23N-2, RGC CRF grants under contracts C7004-22G, C1029-22G and C6015-23G, NSFC project under contract 62432008, and RGC GRF grants under contracts 16207922 and 16207423.

## Impact Statement

The findings of this work have significant implications for the future development and deployment of LLMs in real-world agentic applications. By systematically evaluating the impact of compression techniques on key agentic capabilities, our work sheds light on the complex trade-offs involved in compressing LLMs while preserving their performance in critical tasks such as planning, reasoning, tool use, and long-context understanding.

The introduction of the Agent Compression Benchmark (ACBench) provides a foundational tool for the evaluation of compressed models in agentic settings, an area that has been largely overlooked in prior research. This benchmark paves the way for more targeted research into optimizing LLM compression for specific agent tasks, helping to bridge the gap between state-of-the-art compression methods and the practical requirements of real-world deployments.

Our comprehensive comparison of quantization and pruning techniques highlights the importance of selecting appropriate compression configurations based on the specific needs of agentic tasks. The novel analytical tools we introduce, including ERank, Top-k Ranking Correlation and Energy, provide new insights into the internal workings of compressed models and enable a more granular understanding of the impact of compression on model performance and decision-making.

This work has the potential to influence the design of more efficient LLMs, allowing for their deployment in resource-constrained environments where computational and memory resources are limited. Furthermore, the ability to maintain agentic capabilities even in compressed models will facilitate the broader adoption of LLMs in practical applications such as robotics, autonomous systems, and interactive AI agents. By providing a systematic framework for evaluating and optimizing compressed LLMs, we hope to contribute to the ongoing efforts to make AI models more accessible and capable in real-world scenarios, paving the way for their integration into a wide range of industries and applications.

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

# Appendix

# A. More Related Works

In this section, we provide more related works for the topics discussed in this paper.

## A.1. LLM Compression

Broadly speaking, LLM compression comprises various techniques such as quantization, pruning, knowledge distillation, low-rank decomposition, and more. These methods aim to reduce the size and computational requirements of large language models without substantially sacrificing their performance. By employing these compression strategies, researchers and practitioners can make LLMs more accessible and deployable in resource-constrained environments, thereby broadening their applicability across different platforms and devices.

**Pruning.** Pruning (Frantar & Alistarh, 2023b; Sun et al., 2024c; Shao et al., 2024; Zhang et al., 2024d; Dong et al., 2024b; Tang et al., 2020; Dong et al., 2024a; Lai et al., 2025) involves systematically removing less important parameters or connections within a large language model (LLM) to reduce its size and computational requirements. This technique helps enhance the model's efficiency without significantly compromising its performance by eliminating redundancies. Pruning can be categorized into several types, including unstructured pruning (Sun et al., 2024c; Shao et al., 2024), which targets individual weights, and structured pruning (Molchanov et al., 2019; Ma et al., 2023; Kim et al., 2024), which removes entire neurons, channels, or layers. Additionally, dynamic pruning (Zhang et al., 2024d; Han et al., 2015) methods adjust the sparsity level during training or inference, allowing for more flexible compression tailored to specific deployment needs. Research has shown that judicious pruning can maintain or even sometimes improve model generalization by removing overfitting parameters.

**Quantization.** Quantization (Park et al., 2024; Frantar & Alistarh, 2022; Lee et al., 2024; Du et al., 2024a; Kim et al., 2023; Lin et al., 2023; Dong et al., 2024c; Gu et al., 2025; Du et al., 2024b; Li et al., 2024f) reduces the precision of the model's weights and activations from high-precision formats (like 32-bit floating-point) to lower-bit representations (such as 8-bit integers). This reduction in numerical precision helps decrease the model's memory footprint and speeds up inference times, making LLMs more deployable in resource-constrained environments. Quantization can be applied in various forms, including uniform quantization, which uses the same scale for all weights, and non-uniform quantization, which allows different scales for different layers or weight groups. Advanced techniques like quantization-aware training (QAT) integrate the quantization process into the training phase, enabling the model to better adapt to lower precision and mitigate performance degradation. Research in post-training quantization (PTQ) explores methods to quantize pre-trained models without retraining, preserving accuracy while achieving significant compression.

**Knowledge Distillation.** Knowledge distillation (Hinton et al., 2015b; Li et al., 2024b; 2023b; Liu et al., 2023c; Tian et al., 2022) is a process where a smaller, student model is trained to replicate the behavior of a larger, teacher model. By transferring the knowledge from the teacher to the student, the resulting compressed model retains much of the performance and capabilities of the original while being more efficient in terms of computation and memory usage. This technique involves training the student model using a combination of the original training data and the outputs (such as logits or probability distributions) produced by the teacher model. Various distillation strategies, including teacher-student learning and self-distillation, have been developed to optimize the efficiency and effectiveness of the process. Recent advancements focus on multi-teacher distillation, where multiple teacher models contribute to the training of a single student model, enhancing the diversity and robustness of the compressed model.

**Low-Rank Decomposition.** Low-rank decomposition techniques (Yuan et al., 2023; Zhang et al., 2024b; Wang et al., 2024e; 2025b) aim to approximate the weight matrices of LLMs with lower-rank matrices, effectively reducing the number of parameters and computational operations required during inference. By decomposing large matrices into products of smaller matrices, these methods exploit the inherent redundancy and low-rank structure present in many neural network architectures. Singular Value Decomposition (SVD) is a widely used technique for low-rank approximation, where a matrix is factorized into three components with reduced dimensionality. Other methods, such as tensor decomposition and matrix factorization, have also been explored to facilitate efficient compression. Low-rank decomposition not only diminishes the model size but also accelerates inference by simplifying the mathematical operations involved, making it a valuable tool for deploying LLMs in real-time applications.

## A.2. LLM-based Agents

Large Language Model (LLM)-based agents like MetaGPT (Hong et al., 2023), AutoGen (Wu et al., 2023), AgentVerse (Chen et al., 2023b), and MegaAgent (Wang et al., 2024b) are sophisticated AI systems that leverage the powerful natural language understanding and generation capabilities of LLMs as their core reasoning engine to perform complex, autonomous tasks through interaction with their environment. These agents typically integrate LLMs with additional components such as planning modules, tool use capabilities, and memory systems to create more versatile and capable autonomous systems. The combination of these elements allows LLM-based agents to not only generate human-like text but also engage in goal-directed behavior, adapt to dynamic environments, and utilize external resources to achieve specified objectives.

LLM-based agents operate by receiving inputs or prompts that define their goals or tasks (Chen et al., 2024a; 2023a; Tesfatsion, 2023; Wen et al., 2024; Chan et al., 2024; Park et al., 2023), processing this information through the LLM to generate plans or actions (Chen et al., 2023a; Wang et al., 2024b;c), and then executing these actions in their environment. For instance, an LLM agent designed for web navigation might interpret user queries, plan a sequence of web interactions to gather information, and utilize browser automation tools to execute these interactions effectively. Similarly, an LLM agent tasked with information gathering can autonomously search, retrieve, and synthesize data from various sources to provide comprehensive insights or summaries.

Recent advancements in LLM-based agents have demonstrated remarkable abilities in diverse applications (Hong et al., 2023; Wu et al., 2023; Chen et al., 2023b; Wang et al., 2024b;c), ranging from automated customer service bots that can handle complex queries to virtual assistants capable of managing schedules, composing emails, and even providing personalized recommendations. In more specialized domains, LLM agents have been employed for scientific research assistance (Jennings et al., 1998; Chen et al., 2024a), coding support (Huang et al., 2023; Zhang et al., 2023a;b), and creative content generation (Chen et al., 2024a; 2023a), showcasing their flexibility and adaptability.

However, the development and deployment of LLM-based agents also raise important considerations regarding reliability (Zheng et al., 2023b; Huang et al., 2024a; Xue et al., 2024; Tonmoy et al., 2024), safety (Andriushchenko et al., 2024; Tonmoy et al., 2024), and ethical implications (Hua et al., 2023; Andriushchenko et al., 2024). Ensuring that these agents act responsibly and ethically requires robust mechanisms for oversight, error correction, and adherence to guidelines that prevent misuse or unintended harmful behaviors. Moreover, the integration of planning and memory systems necessitates careful design to maintain data privacy, security, and compliance with regulatory standards.

Furthermore, performance optimization and resource management (Dasgupta et al., 2023; Samsi et al., 2023a; Zhang et al., 2024a; Xia et al., 2023; Liu et al., 2023a) are critical for the efficient functioning of LLM-based agents, especially when deployed in environments with limited computational resources (Samsi et al., 2023a; Tang et al., 2019; Stojkovic et al., 2024; You et al., 2022). Techniques such as distributed computing (You et al., 2022; Wilkins et al., 2024), edge computing (Babakniya et al., 2023), token optimization (Liu et al., 2023a; Xue et al., 2024; Liu et al., 2024b; Zhang et al., 2024a), and model compression (Dong et al., 2024b;c; Hinton et al., 2015a) strategies play a pivotal role in enhancing the scalability and responsiveness of these agents, thereby facilitating their broader adoption across various industries and applications.

## A.3. Reasoning

Reasoning capabilities (Paul et al., 2023; Wei et al., 2022b; Wang et al., 2024c; 2022; Yao et al., 2024; Besta et al., 2024; Luo et al., 2023; Zhou et al., 2023; Zhang et al., 2023d; Xu et al., 2023) in LLMs refer to their ability to process information logically, make inferences, and arrive at conclusions through structured thought processes (Jiang et al., 2023b; Sun et al., 2023; Zhou et al., 2023; Han et al., 2022; Sun et al., 2024a). This encompasses various forms of reasoning, including deductive reasoning (drawing specific conclusions from general principles) (Han et al., 2022; Pan et al., 2023), inductive reasoning (forming generalizations based on specific observations) (Sun et al., 2024a; Xu et al., 2024; 2023), and analogical reasoning (applying knowledge from familiar situations to novel ones) (Xu et al., 2024; 2023; Amirizaniani et al., 2024). The sophistication of reasoning in LLMs is primarily facilitated by their extensive training on diverse textual data, which enables them to recognize patterns, understand contexts, and simulate logical progression (Han et al., 2022; Pan et al., 2023; Xu et al., 2024).

Deductive reasoning (Xu et al., 2023) in LLMs allows them to apply established rules or premises to derive specific outcomes. For example, given the premises "All humans are mortal" and "Socrates is a human," an LLM can deduce that "Socrates is mortal." Inductive reasoning (Sun et al., 2024a; Xu et al., 2024; 2023) enables LLMs to make generalized statements

based on specific instances, such as inferring that "The sun will rise tomorrow" based on past observations. Analogical reasoning (Xu et al., 2024; 2023; Amirizaniani et al., 2024) involves drawing parallels between different scenarios, aiding LLMs in transferring knowledge from one domain to another, which is particularly useful in problem-solving and creative tasks.

Recent research has demonstrated that LLMs can showcase sophisticated reasoning abilities through techniques like chain-of-thought prompting (Wei et al., 2022a; Yao et al., 2024; Besta et al., 2024), where the model generates intermediate reasoning steps that lead to the final answer. This not only improves the transparency of the model's decision-making process but also enhances the accuracy of its responses. Self-consistency (Wang et al., 2023c;b; 2022) checking (Koa et al., 2024; Shinn et al., 2024; Madaan et al., 2024; Li et al., 2024g; 2023c), another technique, involves generating multiple reasoning paths (Leblond et al., 2021; Chakraborty et al., 2024) and selecting the most consistent or probable one, thereby mitigating errors and improving reliability.

However, challenges persist in ensuring that LLMs maintain reliable and consistent logical reasoning across different contexts and domains. Factors such as ambiguous inputs (Lee & Tiwari, 2024; Lu et al., 2023; Gehman et al., 2020), lack of real-world grounding (Zheng et al., 2023a; Xie et al., 2024; Qin et al., 2023b), and limitations in factual knowledge (Li et al., 2024a; Chen et al., 2022; Sachdeva et al., 2024) can impact the quality of reasoning. To address these issues, ongoing research focuses on integrating structured reasoning frameworks (Sun et al., 2023; Luo et al., 2023; Jiang et al., 2023b), enhancing model architectures to better capture logical relationships (Xu et al., 2024), and incorporating external knowledge sources (Jeong et al., 2024; Soudani et al., 2024; Gao et al., 2023a; Chen et al., 2024b; Asai et al., 2024; Wang et al., 2023d; Yu et al., 2024) that provide factual accuracy and context.

Moreover, the interpretability of reasoning in LLMs is a critical area of interest. Developing methods to visualize and understand the internal reasoning processes of these models can lead to better insights into their decision-making mechanisms, facilitating improvements in both performance and trustworthiness. Ensuring that LLMs can reason effectively and transparently is essential for their application in high-stakes environments such as healthcare, law, and scientific research, where accurate and explainable reasoning is paramount.

### A.4. Planning

Planning capabilities (Zhou et al., 2023; Chen et al., 2023a; Song et al., 2023; Rivera et al., 2024; Valmeekam et al., 2022; Zhang et al., 2023c) in LLMs involve the ability to break down complex tasks into manageable steps and determine appropriate sequences of actions to achieve specific goals. This includes both high-level strategic planning and more detailed tactical planning, enabling LLMs to approach problems systematically and execute multi-step processes effectively. The integration of planning mechanisms enhances the autonomy and efficiency of LLM-based systems, making them more robust and adaptable to a variety of applications.

Recent advances have shown that LLMs can engage in various forms of planning, from simple task decomposition (Li et al., 2024c; Wang et al., 2024b), where a task is broken down into subtasks, to more complex hierarchical planning (Zhou et al., 2023; Wang et al., 2024b) approaches that involve multiple layers of strategy and execution. Techniques like tree-of-thought reasoning (Zhou et al., 2023) allow LLMs to explore different branches of possible actions and evaluate their outcomes, fostering more informed decision-making (Xie et al., 2024). Recursive planning enables models to refine and adapt their plans based on intermediate results or feedback, enhancing their ability to handle dynamic and uncertain environments.

Applications of planning in LLMs span various domains, including robotics (Song et al., 2023; Wang et al., 2023a; Zhang et al., 2024c), where autonomous agents must navigate and interact with physical environments; software development (Huang et al., 2024b), where complex coding tasks require structured approaches (Zhang et al., 2023c); and creative industries (Xie et al., 2024), where planning is essential for content creation and project coordination. In each of these areas, the ability to plan effectively enables LLMs to execute tasks with greater precision, efficiency, and reliability.

Continuous research and development are focused on refining planning algorithms (Hong et al., 2024; Zhou et al., 2023; Ge et al., 2023), integrating advanced cognitive frameworks, and leveraging hybrid models that combine symbolic reasoning (Pan et al., 2023; Wang et al., 2024d) with neural network-based approaches.

### A.5. Tool Use

Tool use capability refers to an LLM's ability to effectively utilize external tools and APIs (Zhang et al., 2023a; Schick et al., 2024; Liu et al., 2023b; Patel et al., 2024; Tang et al., 2023a; Shi et al., 2024; Yang et al., 2023b; Qin et al., 2023b) to

accomplish tasks beyond its native capabilities. This includes interfacing with web browsers, calculators, databases, and other specialized software tools. By integrating external tools, LLMs can extend their functionalities, enabling them to perform complex computational tasks, access up-to-date information, and interact with various systems seamlessly.

External tool integration involves several key components:

**Selection of Appropriate Tools.** LLMs must be capable of determining which tools are best suited for a given task. This involves understanding the requirements of the task and matching them with the functionalities offered by available tools. For instance, a task requiring numerical computations might prompt the use of a calculator API, while a task involving data retrieval might utilize a database query tool (Qin et al., 2023b; Shi et al., 2024; Schick et al., 2024; Liu et al., 2023b; Zhang et al., 2023a).

**Constructing Valid API Calls.** Once an appropriate tool is identified, the LLM must be able to construct valid API calls to interact with the tool. This requires understanding the syntax and parameters of the API, as well as the ability to format requests correctly (Qin et al., 2023b; Tang et al., 2023a; Shi et al., 2024). Proper API call construction ensures that the tool is utilized effectively and returns the desired outcomes.

**Interpreting Tool Outputs.** After executing an API call, the LLM needs to interpret the output provided by the external tool. This involves parsing the returned data, understanding its structure, and integrating it with the ongoing task or response generation (Zhang et al., 2023a; Schick et al., 2023). Effective interpretation enables the LLM to provide coherent and contextually appropriate results based on the tool's output.

Recent developments have shown significant progress in enhancing LLMs' tool use capabilities. Techniques such as tool-specific fine-tuning (Schick et al., 2024; Liu et al., 2023b; Tang et al., 2023a; Shi et al., 2024), where models are trained on the usage patterns of specific tools, have improved the accuracy and reliability of tool interactions. Additionally, the incorporation of contextual understanding allows LLMs to dynamically select and switch between tools based on the task requirements and the context in which the task is being performed.

Applications of tool use in LLMs are diverse and impactful (Zhang et al., 2023a; Schick et al., 2024; Qin et al., 2023b). In customer service, LLMs equipped with tool use capabilities can access knowledge bases, perform troubleshooting steps, and process transactions autonomously, providing efficient and effective support. In data analysis, LLMs can interface with analytical tools and databases to perform complex queries, generate reports, and derive insights from large datasets. In creative industries, tool integration enables LLMs to interact with content creation software (Zhang et al., 2023a), facilitating tasks such as graphic design, video editing, and multimedia production (Liu et al., 2023b).

Furthermore, the ability to utilize tools enhances the problem-solving capabilities of LLMs, allowing them to address a broader range of challenges by leveraging specialized functionalities that are beyond their inherent design (Shi et al., 2024; Schick et al., 2024; Qin et al., 2023b). This synergy between LLMs and external tools leads to more robust, versatile, and capable AI systems that can perform intricate tasks with higher accuracy and efficiency.

### A.6. Retrieval Augmented Generation

Retrieval-Augmented Generation (RAG) is an advanced technique that enhances the capabilities of large language models (LLMs) by integrating external knowledge sources during the generation process. Unlike traditional LLMs that rely solely on the information encoded within their parameters, RAG leverages a retrieval component to fetch relevant documents or data from expansive databases or knowledge bases (Jeong et al., 2024; Soudani et al., 2024; Gao et al., 2023a; Chen et al., 2024b; Asai et al., 2024; Wang et al., 2023d; Yu et al., 2024). This approach addresses the limitations of fixed-parameter models, enabling them to access up-to-date and domain-specific information dynamically.

The architecture of RAG typically comprises two primary components: a retriever and a generator (Jeong et al., 2024; Wang et al., 2023d). The retriever utilizes dense or sparse retrieval methods to identify and extract pertinent information related to a given query or prompt from the external corpus. Techniques such as dense vector representations (Pan et al., 2024), which encode semantic similarities, or traditional keyword-based approaches, can be employed to optimize the retrieval process. Once the relevant documents are retrieved, the generator, often powered by transformer-based architectures, assimilates this information to produce coherent and contextually accurate responses.

One of the significant advantages of RAG is its ability to mitigate issues related to factual inaccuracies and hallucinations commonly observed in generative models. By grounding the generation process in retrieved factual data, RAG models can produce more reliable and verifiable outputs. This is particularly beneficial for applications in areas such as healthcare, legal,

and scientific research, where precision and accuracy are paramount.

Recent advancements in RAG have focused on improving the integration between the retrieval and generation stages (Gao et al., 2023b). Innovations such as end-to-end training frameworks allow the retriever and generator to be optimized jointly, enhancing the overall performance and coherence of the generated content. Additionally, scalability challenges are being addressed by developing more efficient retrieval mechanisms that can handle vast and diverse datasets without compromising speed or accuracy.

Applications of Retrieval-Augmented Generation are diverse and rapidly expanding. In customer service, RAG-powered chatbots can provide more informed and relevant responses by accessing a company's knowledge base in real-time. In academic research (Wang et al., 2023d; Zhang et al., 2023b; Gao et al., 2023b), RAG can assist in literature reviews by retrieving and summarizing pertinent studies (Pipitone & Alami, 2024). Furthermore, in creative industries, such as content creation (Wu et al., 2024a) and journalism (Wang et al., 2023d), RAG can aid in generating well-informed and contextually rich narratives (Wu et al., 2024a).

### A.7. Long Context

Processing and understanding long contexts is a formidable challenge in the realm of large language models (LLMs). Traditional transformer architectures, while effective in handling short to moderately long sequences, face scalability issues as the input length increases. This limitation arises primarily due to the quadratic complexity of self-attention mechanisms, which hampers efficiency and computational feasibility for extended texts.

To overcome these challenges, several innovative approaches and architectural modifications have been proposed to enhance the long-context capabilities of LLMs:

**Sparse Attention Mechanisms**    Sparse attention reduces the computational burden by limiting the scope of attention to a subset of tokens rather than considering all possible token pairs. Techniques such as Longformer's sliding window attention and BigBird's combination of global and random attention patterns enable models to process longer sequences efficiently while maintaining performance on downstream tasks.

**Memory-Augmented Networks**    Memory-augmented models incorporate external memory structures that allow the model to store and retrieve information across longer spans. Transformer-XL, for instance, introduces a recurrence mechanism that reuses hidden states from previous segments, effectively extending the context window without a proportional increase in computational resources.

**Hierarchical Models**    Hierarchical approaches decompose the processing of long texts into multiple levels of abstraction. By first encoding smaller chunks of text (e.g., sentences or paragraphs) and then aggregating these encodings at a higher level, models can capture long-range dependencies more effectively. This method not only enhances comprehension but also facilitates better information retention across lengthy documents.

**Receptive Field Modifications**    Adjusting the receptive field of the attention mechanism allows models to maintain a broadened scope of context without incurring significant computational costs. Techniques like dynamic attention span, where the model learns the optimal span of attention based on the input, contribute to more flexible and efficient long-context processing.

**Efficient Positional Encoding**    Positional encodings play a crucial role in maintaining the order of tokens in sequences. For long contexts, standard positional encodings become less effective. Innovations such as relative positional encodings and rotary embeddings help preserve positional information over extended sequences, ensuring that the model retains a coherent understanding of the text structure.

**Segment-Based Processing**    Breaking down long documents into manageable segments and processing them iteratively is another strategy to handle long contexts. This approach involves processing each segment independently while maintaining a summary or representation that captures the essential information from previous segments. Models like Reformer employ such techniques to sustain context over extended inputs.

**Adaptive Computation**    Adaptive computation strategies allow models to allocate computational resources dynamically based on the complexity and relevance of different parts of the input. By focusing more on critical segments and allocating fewer resources to less pertinent sections, models can efficiently manage long contexts without compromising performance.

The ability to effectively handle long contexts is pivotal for numerous applications, including document summarization (Jiang, 2024), in-depth question answering, narrative generation, and complex dialogue systems. Enhancing long-context processing not only improves the performance of LLMs on tasks requiring extended reasoning and memory but also expands their applicability across diverse domains where understanding large volumes of information is essential.

Continuous research and development in this area are driving the evolution of more sophisticated models that can seamlessly integrate and interpret long-range dependencies, ultimately leading to more intelligent and contextually aware AI systems.

## B. Detailed Experiment Settings

### B.1. Details of Quantization and Sparsification

For our quantization and sparsification experiments, we evaluated several state-of-the-art compression methods across different model architectures:

**Quantization Methods**    We employed two primary quantization approaches: (1) AWQ (Activation-aware Weight Quantization) (Lin et al., 2023), which adaptively determines quantization parameters based on activation patterns during inference; and (2) GPTQ (Frantar et al., 2022), a post-training quantization method that uses second-order information to optimize quantization parameters. We employed the LLMC (Gong et al., 2024) framework for unified quantization and aligned the settings in LLMC.

**Sparsification Methods**    We investigated both structured and unstructured pruning approaches: (1) Magnitude Pruning, both unstructured (Mag(Un)) and structured 2:4 (Mag(2:4)) variants, which remove weights based on their absolute values; (2) Wanda (Sun et al., 2024b), applied in both unstructured (Wanda(Un)) and structured 2:4 (Wanda(2:4)) configurations, offering layer-wise adaptive pruning; and (3) SparseGPT (Frantar & Alistarh, 2023a), implemented in unstructured (SparseGPT(Un)) and structured 2:4 (SparseGPT(2:4)) formats, using gradient information for pruning decisions. We employed Wanda (Sun et al., 2024b) as the repository for Magnitude, Wanda, and SparseGPT.

All compression methods were applied to pre-trained models without additional fine-tuning. For structured pruning, we maintained the 2:4 sparsity pattern (50% sparsity) across all layers. Unstructured pruning targeted similar overall sparsity levels but allowed for flexible weight removal patterns.

### B.2. Details of Small Language Models and Distilled Models

We evaluated several categories of compressed models:

**Small Language Models**    (1) Qwen2.5-3B (Yang et al., 2024b), a smaller variant of Qwen, evaluated with both AWQ and GPTQ quantization; (2) Megrez-3B, a 3B parameter model showing strong performance on multi-doc QA tasks; (3) MiniCPM-4B (Hu et al., 2024), a compact model optimized for efficiency; (4) Gemma-2B (Team, 2024b), Google's lightweight model focused on general-purpose tasks; and (5) Phi-3.5 (Li et al., 2023d), Microsoft's small-scale model emphasizing reasoning capabilities.

**Distilled Models**    We examined three DeepSeek distilled models (Team, 2024a): (1) DeepSeek-R1-Distill-LLama3.1-8B, a distilled version of LLaMA, showing moderate performance on few-shot learning tasks; (2) DeepSeek-R1-Distill-Qwen2.5-1.5B, a heavily compressed version of Qwen, optimized for efficiency; and (3) DeepSeek-R1-Distill-Qwen2.5-7B, a larger distilled variant maintaining better performance than its smaller counterpart. All models were evaluated across various tasks, including long-context understanding, tool use, and real-world applications, to assess their capabilities and limitations under different scenarios.

### B.3. Details of Tool Use Experiments

We mainly employ the T-Eval (Chen et al., 2023c) to conduct experiments. In this dataset, 15 tools were carefully selected from domains such as Research, Travel, Entertainment, Web, Life, and Financials, ensuring high availability and usage

rate, as well as complete documentation. The dataset includes 553 high-quality query-solution annotation pairs, resulting in a total of 23,305 test cases across the Instruct, Plan, Reason, Retrieve, Understand, and Review subsets. The dataset is designed to provide a comprehensive and fine-grained evaluation of tool utilization capabilities, with an average of 5.8 tool calling steps per query, ensuring generalization and discrimination for tool utilization evaluation.

### B.4. Details of Long Context Understanding Experiments

**LongBench:** LongBench (Bai et al., 2024) is a comprehensive benchmark for evaluating the long-context capabilities of LLMs. It consists of 12 tasks across five categories: Single-Doc QA, Multi-Doc QA, Summarization, Few-shot Learning, and Synthetic tasks. The benchmark features documents ranging from 2K to 32K tokens in length. Single-Doc QA tasks include NarrativeQA and Qasper, which test comprehension of long narratives and scientific papers. Multi-Doc QA encompasses HotpotQA and MultiNews, evaluating cross-document reasoning. The summarization tasks (e.g., GovReport, QMSum) assess models' ability to distill key information from lengthy documents. Few-shot learning tasks test the model's capacity to learn from examples within extended contexts, while synthetic tasks measure specific capabilities like information retrieval and counting in long sequences. The benchmark spans diverse domains including healthcare, legal, scientific research, and creative writing, providing a comprehensive evaluation of long-context understanding.

**LongGenBench:** LongGenBench (Liu et al., 2024a) is a synthetic benchmark designed to evaluate long-context generation capabilities of language models. Unlike existing long-context benchmarks that focus primarily on retrieval-based tasks (e.g., locating specific information within large input contexts), LongGenBench specifically targets a model's ability to produce coherent, contextually accurate, and structurally sound text over extended outputs. The benchmark supports customizable context length configurations and evaluates models on generating single, unified long-form responses. This fills an important gap in long-context evaluation by providing a standardized framework for assessing generation rather than just retrieval capabilities. The benchmark enables systematic analysis of how different model architectures handle the challenges of maintaining coherence and accuracy in long-form text generation.

**Needle-in-the-haystack:** This benchmark (Kamradt, 2023) specifically tests models' ability to locate and utilize critical information embedded within lengthy contexts. The test suite includes documents ranging from 4K to 64K tokens, with key information deliberately placed at varying positions (beginning, middle, end) and with different levels of surrounding distractor content. Tasks include fact retrieval, where specific details must be located within long documents; reasoning chains, where multiple pieces of information must be connected across the document; and verification tasks, which test the model's ability to find evidence supporting or contradicting given claims. The benchmark employs various document structures (narrative, technical, dialogue) and information patterns (explicit statements, implicit connections, numerical data) to comprehensively evaluate information retrieval capabilities in long contexts. Performance is measured not only on accuracy but also on efficiency in identifying relevant information without being misled by distractors.

### B.5. Details of Workflow Generation Experiments

In this paper, we employ WorfBench (Qiao et al., 2024) to evaluate the workflow generation capabilities, which encompass planning and reasoning processes. WorfBench is a comprehensive benchmark containing 18k training samples, 2146 test samples, and 723 held-out tasks across four types of scenarios: function call, problem-solving, embodied planning, and open-grounded planning. The benchmark features multi-faceted scenarios and complex graph-structured workflows, addressing limitations of existing benchmarks that focus only on linear workflows. WorfBench is accompanied by WorfEval, a systematic evaluation protocol that uses subsequence and subgraph matching algorithms to accurately quantify workflow generation abilities. This evaluation framework has revealed significant gaps between linear and graph planning capabilities in current LLMs, with even advanced models showing up to 15% performance differences. The benchmark's diverse scenarios and evaluation metrics make it particularly suitable for assessing both the quality of generated workflows and their practical applicability in real-world tasks.

### B.6. Details of Real-World Applications Experiments

In this paper, we employ the AgentBoard (Ma et al., 2024) to evaluate real-world applications. The benchmark is designed to comprehensively evaluate large language models (LLMs) as generalist agents, comprising a diverse set of 9 unique tasks and 1013 environments. These tasks cover a wide range of scenarios, including embodied AI tasks (AlfWorld, ScienceWorld, BabyAI), game environments (Jericho, PDDL), web-based tasks (WebShop, WebArena), and tool-oriented tasks (Tool-Query, Tool-Operation). Each environment is carefully crafted to ensure multi-round interactions and partially

Figure 9: Performance comparison of sparse compression methods on tool use tasks

observable characteristics, with subgoals defined or annotated to track detailed progress. The benchmark also introduces a fine-grained progress rate metric to capture incremental advancements, providing a more nuanced evaluation beyond just the final success rate. Additionally, AgentBoard offers an open-source evaluation framework with an interactive visualization panel to support detailed analysis of agent abilities across various dimensions.

## C. More Experiment Results

### C.1. Long-Context Understanding

Table 3 presents a comparative analysis of the AWQ and GPTQ quantization methods applied to two large language models (LLMs), InternLM2.5-7B and Qwen2.5-7B, across various long-context understanding tasks sourced from LongBench.

For InternLM2.5-7B, the application of the AWQ results in significant performance enhancements across most evaluated tasks. Specifically, in Single-Doc QA tasks such as NrtvQA, Qasper, and MF-en, AWQ boosts the scores from 0 to 25.08, 29.27 to 45.16, and 22.92 to 47.72, respectively. Similarly, in Multi-Doc QA and Summarization tasks, AWQ consistently improves performance metrics, with notable increases in TriviaQA (from 48.1 to 84.69) and Code Completion (from 50 to 99.5). In contrast, the GPTQ method yields more modest improvements. While GPTQ enhances some scores, such as Qasper from 29.27 to 30.16 and MF-en from 22.92 to 30.04, the gains are less pronounced compared to AWQ. Additionally, GPTQ shows slight decreases in certain areas, such as Lcc, which drops from 58.2 to 56.62.

Examining Qwen2.5-7B, both AWQ and GPTQ quantization methods demonstrate varying levels of performance enhancement. AWQ provides consistent, albeit smaller, improvements across most tasks. For example, Qasper increases from 13.12 to 14.18, and MF-en decreases slightly from 30.29 to 26.12. On the other hand, GPTQ offers substantial performance gains in several areas, notably increasing Qasper from 13.12 to 38.94 and MF-en from 30.29 to 47.53. However, similar to InternLM2.5-7B, GPTQ also results in slight performance declines in specific tasks, such as a reduction in Lcc from 62.28 to 57.44.

Overall, AWQ tends to provide more stable and consistent improvements across a broad range of tasks for both models, making it a reliable choice for enhancing long-context understanding capabilities. GPTQ, while offering significant boosts in certain key areas, presents more variability in performance outcomes. These findings suggest that the choice between AWQ and GPTQ should be informed by the specific performance requirements and the nature of the tasks at hand.

Table 4 presents a detailed comparison of various sparsification methods applied to InternLM2.5-7B and Qwen2.5-7B across different long-context understanding tasks from LongBench. For InternLM2.5-7B, unstructured sparsification methods consistently outperform their structured (2:4) counterparts across most tasks. Among the unstructured approaches,

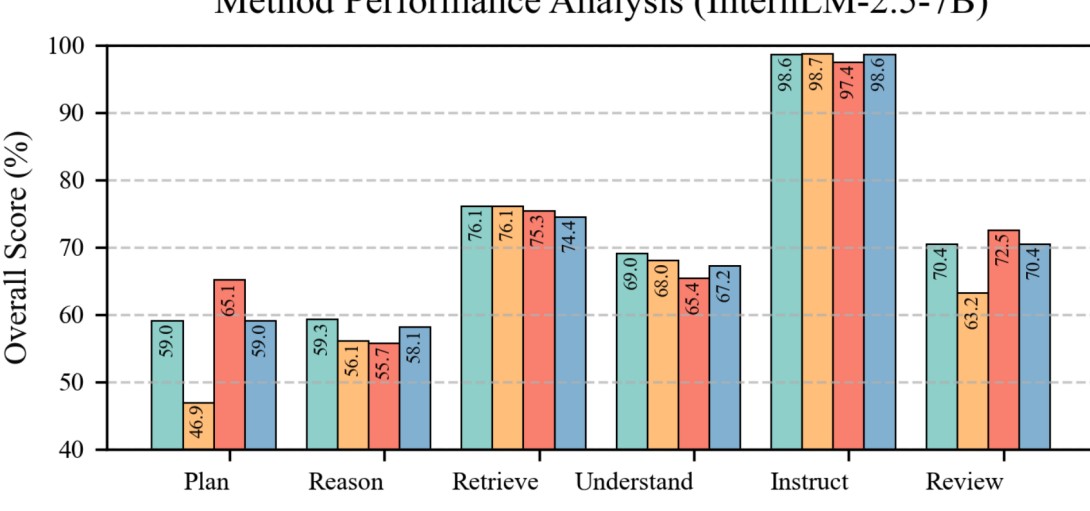

Figure 10: Performance comparison of quantization methods on tool use tasks

Table 3: Performance Comparison of AWQ and GPTQ Quantization Methods on Long-Context Understanding Tasks from LongBench

| LLMs | Quantization | Single-Doc QA | | | Multi-Doc QA | | | | Summarization | | | | Few-shot Learning | | | Synthetic Task | | Code Completion |
|---|---|---|---|---|---|---|---|---|---|---|---|---|---|---|---|---|---|---|
| | | NrtvQA | Qasper | MF-en | HotpotQA | 2WikiMQA | Musique | Dureader | GovReport | QMSum | MultiNews | VCSum | TREC | TriviaQA | SAMSum | LSHT | PRE | Pcount | Lcc |
| InternLM2.5-7B | - | 0 | 29.27 | 22.92 | 4.43 | 22.51 | 15.26 | 12.8 | 16.04 | 12.15 | 24.03 | 9.82 | 41.5 | 48.1 | 18.53 | 21.75 | 50 | 0 | 58.2 |
| | AWQ | 25.08 | 45.16 | 47.72 | 36.52 | 36.18 | 25.6 | 26.26 | 28.64 | 24.17 | 25.82 | 17.21 | 69.5 | 84.69 | 30.88 | 42.5 | 99.5 | 0 | 61.68 |
| | GPTQ | 12.29 | 30.16 | 30.04 | 18.53 | 37.22 | 21.75 | 25.84 | 16.51 | 11.98 | 24.21 | 10.03 | 40.75 | 56.04 | 17.52 | 20.75 | 50 | 0.05 | 56.62 |
| Qwen2.5-7B | - | 7.99 | 13.12 | 30.29 | 10.78 | 10.42 | | 32.32 | 33.96 | 21.85 | 24.93 | 17.3 | 74 | 85.17 | 49.48 | 42.5 | 98.67 | | 62.28 |
| | AWQ | 8.4 | 14.18 | 26.12 | 11.84 | 10.95 | 7.36 | 33.3 | 33.16 | 21.72 | 24.57 | 17.26 | 74 | 87.73 | 46.17 | 41.75 | 93.75 | 1.43 | 59.52 |
| | GPTQ | 28.34 | 38.94 | 47.53 | 54.53 | 38.88 | 21.73 | 30.77 | 33.31 | 24.16 | 25.84 | 18.26 | 72.5 | 89.04 | 45.45 | 42.08 | 95 | 8 | 57.44 |

Wanda(un) demonstrates superior performance, achieving the highest scores in several categories including NrtvQA (23.94), Qasper (41.21), and MF-en (45.07). The model also maintains strong performance in synthetic tasks and code completion, with scores of 99.5 for PRE and 58.49 for Lcc respectively. SparseGPT(un) follows closely behind, showing particularly strong results in Multi-Doc QA tasks.

For Qwen2.5-7B, the performance pattern differs notably. Magnitude pruning, both structured and unstructured, shows significant degradation across most tasks, with scores dropping to near-zero in many cases. However, SparseGPT and Wanda methods maintain relatively robust performance. Wanda(un) achieves the best overall performance, particularly in Single-Doc QA tasks (16.3 for NrtvQA) and code completion metrics (45.14 for Lcc). SparseGPT(un) also demonstrates strong performance, especially in few-shot learning tasks, achieving 89.35 on TriviaQA and maintaining high scores on synthetic tasks (92.54 on PRE).

Table 5 presents a comprehensive evaluation of small language models and distilled DeepSeek models on LongBench. Among the small models, Qwen-3B demonstrates strong performance across most tasks, particularly when quantized with GPTQ, maintaining comparable performance to its base model. For instance, GPTQ-quantized Qwen-3B achieves 21.39 on NrtvQA and 49.29 on MF-en, showing minimal degradation from the base model's 22.05 and 49.6 respectively. However, AWQ significantly impacts its performance, with near-zero scores on several tasks.

Megrez-3B (Infinigence, 2024) stands out among small models with exceptional performance on multi-doc QA tasks (67.62 on HotpotQA, 57.7 on 2WikiMQA) and few-shot learning (82.5 on TREC). In contrast, newer models like MiniCPM-4B, Gemma-2B, and Phi-3.5 show limited capabilities on long-context tasks, particularly struggling with single-doc and multi-doc QA tasks. The distilled models (DeepSeek-LLama-8B, DeepSeek-Qwen-1.5B, DeepSeek-Qwen-7B) generally show moderate performance, with DeepSeek-LLama-8B performing better on few-shot learning tasks (87.55 on TriviaQA) but all showing significant degradation in QA and summarization tasks compared to their teacher models.

Table 4: Performance Comparison of Sparsification Methods Across Different Tasks for InternLM2.5-7B and Qwen2.5-7B on LongBench

| LLMs | Sparification | Single-Doc QA | | | | Multi-Doc QA | | | Summarization | | | | Few-shot Learning | | | | Synthetic Task | | Code Completion | |
|---|---|---|---|---|---|---|---|---|---|---|---|---|---|---|---|---|---|---|---|---|
| | | NrtvQA | Qasper | MF-en | HotpotQA | 2WikiMQA | Musique | Dureader | GovReport | QMSum | MultiNews | VCSum | TREC | TriviaQA | SAMSum | LSHT | PRE | Pcount | Lcc | RB-P |
| InternLM2.5-7B | Mag(2:4) | 8.01 | 17.88 | 33.58 | 24.72 | 23.35 | 12.42 | 15.62 | 16.31 | 21.13 | 21.47 | 6.74 | 45.5 | 58.54 | 30.11 | 20.5 | 66.85 | 3.96 | 26.76 | 38.06 |
| | Mag(un) | 18.49 | 23.11 | 39.73 | 31.59 | 25.29 | 19.37 | 20.67 | 28.44 | 25.39 | 26.2 | 13.59 | 60.5 | 79.26 | 29.28 | 27.5 | 96.25 | 3.33 | 49.72 | 45.66 |
| | SparseGPT(2:4) | 9.58 | 23.01 | 29.03 | 15.39 | 26.57 | 13.23 | 11.55 | 19.03 | 16.72 | 23.12 | 10.75 | 55.33 | 70.84 | 26.35 | 41.5 | | | | |
| | SparseGPT(un) | 21.16 | 40.13 | 44.44 | 34.21 | 35.69 | 21.22 | 26.78 | 30.85 | 24.74 | 26.11 | 16.57 | 65 | 84.43 | 22.49 | 42.5 | 99 | 0.45 | 49.42 | 49.31 |
| | Wanda(2:4) | 19.34 | 34.45 | 44.19 | 25.98 | 25.61 | 13.53 | 27.6 | 27.49 | 25.42 | 24.59 | 14.8 | 67.5 | 83.55 | 31.63 | 42.25 | 99 | 0.5 | 31.45 | 41.53 |
| | Wanda(un) | 23.94 | 41.21 | 45.07 | 32.65 | 34.06 | 20.1 | 29.63 | 29.61 | 25.23 | 25.87 | 18.05 | 65 | 87.44 | 23.75 | 41 | 99.5 | 0.5 | 58.49 | 51.94 |
| Qwen2.5-7B | Mag(2:4) | 0.72 | 1.08 | 3.25 | 2.43 | 2.49 | 1.48 | 8.01 | 1.85 | 1.9 | 3.23 | 3.65 | 16 | 9.21 | 2.92 | 14 | 0.25 | 0 | 12.36 | 12.01 |
| | Mag(un) | 0.84 | 0.28 | 4.77 | 1.81 | 2.95 | 0.53 | 9.89 | 1.43 | 1.72 | 2.01 | 11.3 | 36.5 | 21.87 | 10.96 | 18 | 0.5 | 0 | 10.6 | 16.59 |
| | SparseGPT(2:4) | 3.36 | 9.16 | 24.45 | 11.2 | 10.92 | 6.3 | 26.1 | 30.5 | 23.45 | 26.16 | 14.63 | 69 | 84.76 | 43.75 | 42 | 75.46 | 5.12 | 30.41 | 35.92 |
| | SparseGPT(un) | 4.95 | 12.59 | 27.35 | 11.82 | 12.27 | 6.39 | 29 | 33.52 | 22.67 | 25.77 | 16.47 | 69.5 | 89.35 | 45.51 | 39 | 92.54 | 3.11 | 39.7 | 42.13 |
| | Wanda(2:4) | 2.21 | 6.32 | 23.28 | 8.91 | 9.27 | 5.4 | 32.43 | 28.61 | 24.01 | 26.06 | 16.17 | 76.5 | 84.39 | 43.1 | 44.5 | 83.23 | 2.46 | 39.34 | 42.19 |
| | Wanda(un) | 16.3 | 11.81 | 29.32 | 10.86 | 11.86 | 7.7 | 33.19 | 32.13 | 24.43 | 26.97 | 17.03 | 75.5 | 88.7 | 46.11 | 41.5 | 93.92 | 1.02 | 45.14 | 46 |

Table 5: Performance Comparison of Small Language Models and Distilled DeepSeek Models on LongBench

| LLMs | Compression | Single-Doc QA | | | | Multi-Doc QA | | | Summarization | | | | Few-shot Learning | | | | Synthetic Task | | Code Completion | |
|---|---|---|---|---|---|---|---|---|---|---|---|---|---|---|---|---|---|---|---|---|
| | | NrtvQA | Qasper | MF-en | HotpotQA | 2WikiMQA | Musique | Dureader | GovReport | QMSum | MultiNews | VCSum | TREC | TriviaQA | SAMSum | LSHT | PRE | Pcount | Lcc | RB-P |
| Qwen-3b | - | 22.05 | 35.12 | 49.6 | 46.82 | 37.33 | 20.5 | 34.92 | 33.71 | 25.44 | 25.3 | 15.66 | | | | | | | | |
| | AWQ | 0 | 10.41 | 13.78 | 1.29 | 3.5 | 0 | 0 | 2.84 | 0.11 | 21.39 | 2.62 | 14.5 | 9.7 | 6.96 | 0 | 0 | 0 | 44.46 | 5.75 |
| | GPTQ | 21.39 | 33.77 | 49.29 | 45.74 | 39.02 | 23.9 | 32.61 | 34.1 | 26.44 | 25.62 | 15.49 | 68.5 | 83.08 | 44.72 | 41 | 23 | 2.5 | 50.41 | 51.47 |
| Qwen-1.5b | - | 21.22 | 37.77 | 49.38 | 41.74 | 32.58 | 23.24 | 30.84 | 30.58 | 24.87 | 25.8 | 15.88 | 74 | 84.08 | 44.02 | 35 | 29.5 | 2.5 | 37.56 | 45.67 |
| | AWQ | 0 | 11.99 | 15.86 | 1.62 | 4.72 | 0 | 0 | 2.79 | 0.06 | 22.26 | 2.69 | 13.5 | 10.16 | 6.26 | 0 | 0 | 0 | 20.22 | 3.2 |
| | GPTQ | 0 | 13.04 | 13.21 | 1.62 | 4.14 | 0 | 0 | 2.62 | 0.08 | 22.33 | 2.67 | 12.5 | 10.56 | 6.53 | 0 | 0 | 0 | 28.14 | 4.84 |
| Megrez-3b | - | 23.7 | 42.74 | 50.45 | 67.62 | 57.7 | 44.31 | 49.43 | 38.33 | 21.81 | 20.08 | 15.44 | 82.5 | 1 | 38.34 | 80 | 94 | 18.5 | 0 | 0.31 |
| MiniCPM-4b | - | 0 | 13.81 | 12.99 | 1.12 | 2.89 | 0 | 0 | 2.23 | 0.12 | 21.01 | 2.85 | 11 | 9.25 | 1.37 | 0 | 0 | 0 | 37.29 | 0.84 |
| Gemma-2b | - | 0 | 21.82 | 20.58 | 4.26 | 13.54 | 0.62 | 0.91 | 7.18 | 1.55 | 22.33 | 2.46 | 28.25 | 18.95 | 8.5 | 0.75 | 0 | 0 | 56.6 | 13.02 |
| Phi-3.5 | - | 0 | 12.74 | 13.26 | 0.91 | 2.77 | 0 | 0.05 | 2.07 | 0.09 | 20.2 | 0.35 | 24.5 | 16.04 | 4.17 | 4.5 | 0 | 0 | 50.97 | 16.18 |
| DS-LLama-8b | Distilled | 3.62 | 9.24 | 12.5 | 2.57 | 7.6 | 1.77 | 16.31 | 27.55 | 17.65 | 22.42 | 11.75 | 67.5 | 87.55 | 42.73 | 44 | 0 | 1.5 | 45.07 | 46.34 |
| DS-Qwen-1.5b | Distilled | 4.54 | 12.32 | 16.29 | 5.77 | 6.3 | 3.73 | 6.5 | 15.02 | 8.54 | 8.04 | 10.95 | 51 | 46.46 | 32.72 | 20.25 | 4.75 | 0.05 | 30 | 35.28 |
| DS-Qwen-7b | Distilled | 3.45 | 9.77 | 11.95 | 2.8 | 7.68 | 2.01 | 12.7 | 26.8 | 16.72 | 22.18 | 13.14 | 66.5 | 78.42 | 35.59 | 23.75 | 1 | 0.83 | 37.92 | 41.79 |

Table 6 presents the performance comparison of various quantization and sparsification methods on LongGenBench. For Qwen2.5-7B, AWQ achieves the best performance among all compression methods, with 52.67 on GSM8K and 64.39 on MMLU, which is very close to the base model's performance (61.33 and 64.65 respectively). However, magnitude-based pruning methods (both unstructured and 2:4) completely fail on these tasks, achieving 0 scores. Wanda and SparseGPT show moderate performance degradation, with unstructured variants generally performing better than 2:4 structured pruning. For InternLM2.5-7B, most compression methods lead to significant performance drops, with only AWQ maintaining reasonable performance (6.50 on GSM8K and 59.12 on MMLU compared to base model's 10.50 and 61.49).

Table 6: Performance Comparison of Quantization and Sparsification Methods on LongGenBench

| LLM | Compression | GSM8K | MMLU | LLM | Compression | GSM8K | MMLU |
|---|---|---|---|---|---|---|---|
| Qwen2.5-7B | Base | 61.33 | 64.65 | InternLM2.5-7B | Base | 10.50 | 61.49 |
| | Mag(Un) | 0.00 | 0.00 | | Mag(Un) | 1.67 | 4.91 |
| | Mag(2:4) | 0.00 | 0.00 | | Mag(2:4) | 0.00 | 0.00 |
| | Wanda(Un) | 35.00 | 57.54 | | Wanda(Un) | 7.00 | 7.02 |
| | Wanda(2:4) | 2.17 | 37.02 | | Wanda(2:4) | 3.33 | 3.33 |
| | SparseGPT(Un) | 18.00 | 58.86 | | SparseGPT(Un) | 6.00 | 10.09 |
| | SparseGPT(2:4) | 7.67 | 43.16 | | SparseGPT(2:4) | 0.17 | 1.23 |
| | GPTQ | 47.50 | 37.28 | | GPTQ | 5.50 | 10.18 |
| | AWQ | 52.67 | 64.39 | | AWQ | 6.50 | 59.12 |

## C.2. Tool Use

In Table 8, we present a comprehensive evaluation of various compression methods across different LLMs on tool use tasks. The results reveal several key findings: (1) InternLM2.5-7B demonstrates strong overall performance, with AWQ achieving the best results (68.6%) among compression methods, closely followed by GPTQ (71.8%) and the base model (71.4%). (2) For Mistral-7B, unstructured pruning methods (Mag(Un), SparseGPT(Un), Wanda(Un)) generally outperform their structured counterparts, though with lower absolute performance compared to InternLM2.5-7B. (3) Qwen2.5-14B shows

Table 7: Performance Comparison of Small Language Models and DeepSeek Distilled Models on LongGenBench

| LLM | Compression | GSM8K | MMLU |
|---|---|---|---|
| Qwen2.5-1.5B | GPTQ(W8) | 3.67 | 32.89 |
| | GPTQ(W4) | 1.17 | 35.61 |
| | AWQ | 2.17 | 16.49 |
| Qwen2.5-3B | GPTQ(W8) | 11.83 | 54.39 |
| | GPTQ(W4) | 11.83 | 50.18 |
| | AWQ | 14.67 | 51.93 |
| Small LM | Phi-3.5 | 3.33 | -1.00 |
| | Megrez-3b | 1.67 | 6.14 |
| Distilled LM | DS-Qwen-7b | 0.00 | 0.00 |
| | DS-Qwen-1.5b | 0.00 | 0.09 |
| | DS-LLama-8b | 3.67 | 2.28 |

impressive resilience to compression, with SparseGPT(Un) maintaining performance close to FP16 (77.0% vs 77.5%). (4) Among smaller models, Qwen2.5-7B exhibits strong performance with AWQ and GPTQ (around 70%), while distilled models generally underperform. (5) Across all models, string format tasks typically see better performance than JSON format, suggesting that compression methods better preserve natural language processing capabilities compared to structured format handling.

### C.3. Real-World Applications

The AgentBoard evaluation results in Table 9 reveal several key insights about model compression performance. For Qwen2.5-7B, AWQ and GPTQ maintain relatively strong performance compared to the base model, with AWQ achieving 19.73% and 47.16% on Jericho and Tool-query tasks, respectively. However, magnitude-based pruning (Mag) shows significant degradation, dropping to near 0% across most tasks. For InternLM2.5-7B, the base model performs moderately with 14.23% on Jericho and 31.64% on Tool-query, but compression methods like Wanda and SparseGPT see notable drops in performance. Distillation approaches (DeepSeek-Qwen-7B, DeepSeek-Qwen-1.5B, DeepSeek-LLama-8B) show very poor performance across all tasks, suggesting that knowledge distillation may not be suitable for these complex agent-based scenarios. Overall, the results indicate that while quantization methods like AWQ and GPTQ can maintain reasonable performance, aggressive pruning and distillation lead to significant capability loss on agent-based tasks.

### C.4. Workflow Generation

The WorfBench evaluation results in Table 10 reveal several key insights about model compression performance. For Qwen2.5-7B, quantization methods like AWQ and GPTQ maintain relatively strong performance with F1 scores around 0.44-0.46, while pruning methods like Wanda and SparseGPT achieve similar F1 scores in the 0.46 range. However, magnitude-based pruning (Mag) shows complete degradation with 0.00 F1 scores across all tasks. Larger models like Qwen2.5-32B demonstrate better compression robustness, with AWQ and GPTQ maintaining F1 scores around 0.69-0.71 compared to the base model's 0.72. Distillation approaches (DeepSeek-R1-Distill models) show significant performance drops with F1 scores between 0.20-0.43. The results suggest that while quantization methods can effectively compress models with minimal performance impact, aggressive pruning and distillation lead to substantial capability loss on workflow generation tasks.

### C.5. Needle-in-the-haystack Test

The needle-in-haystack test results reveal several key insights about model compression methods and their impact on long-range attention capabilities. For InternLM-2.5-7B (Figure 11), we observe that while GPTQ and AWQ maintain relatively stable attention patterns similar to the base model, Wanda and SparseGPT exhibit some degradation in capturing long-range dependencies, particularly beyond the 20k token range. Magnitude pruning shows the most severe impact, with significant attention pattern distortion. For Qwen-2.5-7B (Figure 12), the results demonstrate better compression robustness,

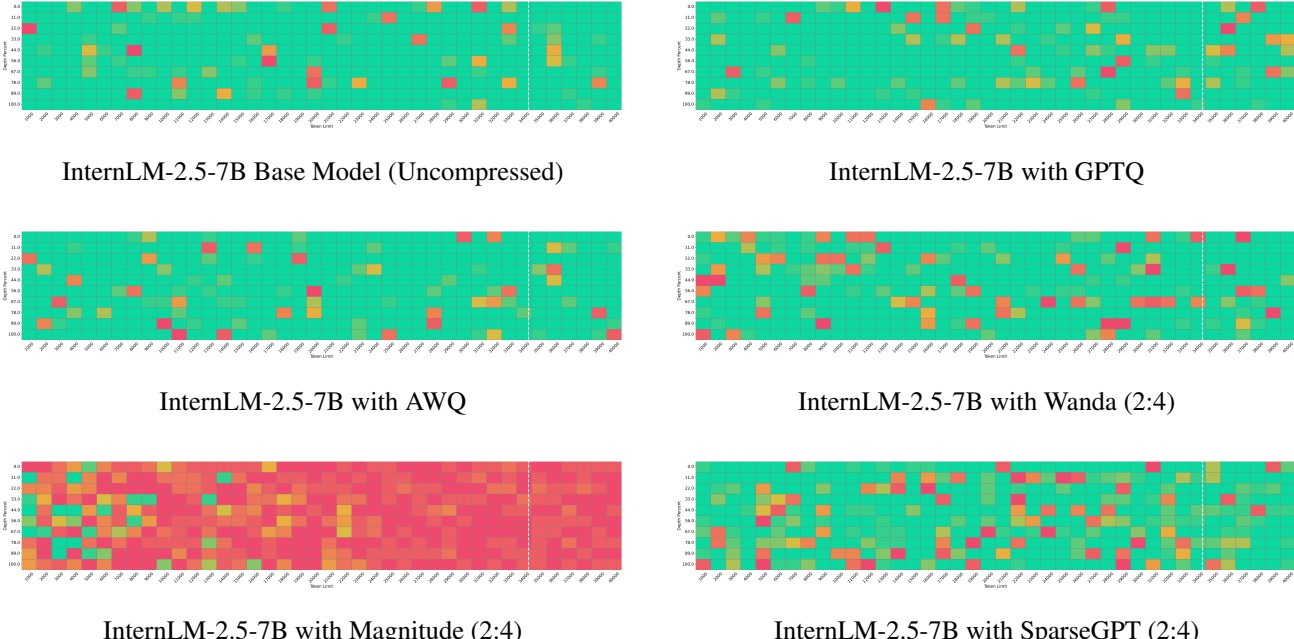

Figure 11: Visualization of attention patterns in needle-in-haystack tests across different compression methods for InternLM-2.5-7B. The heatmaps demonstrate how various compression techniques affect the model's ability to maintain attention over long sequences (40k tokens), with darker colors indicating stronger attention weights.

with both GPTQ and AWQ preserving attention patterns remarkably well across the full 40k context. However, structured pruning methods like Magnitude and SparseGPT still show noticeable degradation in long-range attention. The smaller and distilled models (Figure 13) generally exhibit weaker long-range attention capabilities compared to their larger counterparts, though DeepSeek-Qwen-7B shows promising results in maintaining attention patterns. These visualizations highlight the trade-offs between model compression and the preservation of long-range attention mechanisms, suggesting that careful selection of compression methods is crucial for maintaining model performance on long-context tasks.

### C.6. Logits Visualization

Figure 14 presents a comprehensive visualization of logits distributions across different sequence positions in compressed models. The analysis reveals how the models' prediction confidence and token probability distributions evolve throughout the sequence. At early positions (token 2), we observe relatively focused distributions, indicating strong initial predictions. As we progress through the sequence (tokens 82 and 134), the distributions become more diverse, suggesting that the models maintain their ability to consider multiple plausible continuations. Notably, even at later positions (tokens 184, 225, and 246), the compressed models demonstrate stable logits patterns, indicating that they preserve their predictive capabilities across longer contexts. This visualization helps validate that model compression techniques maintain coherent probability distributions across sequence lengths.

### C.7. Energy Visualization

Figure 15 illustrates the energy distributions across different sequence positions in compressed models. The visualization reveals interesting patterns in how the models' energy landscapes evolve throughout token generation. At the initial position (token 1), we observe a relatively concentrated energy distribution, suggesting focused model attention. As the sequence progresses through intermediate positions (tokens 64 and 117), the energy distributions show increased spread, indicating the models maintain flexibility in considering multiple possible token paths. At later positions (tokens 175, 244, and 252), the energy patterns remain stable and well-structured, demonstrating that the compressed models preserve their energy-based decision-making capabilities even in longer sequences. This analysis validates that our compression techniques successfully maintain the models' ability to capture meaningful energy landscapes across different sequence lengths.

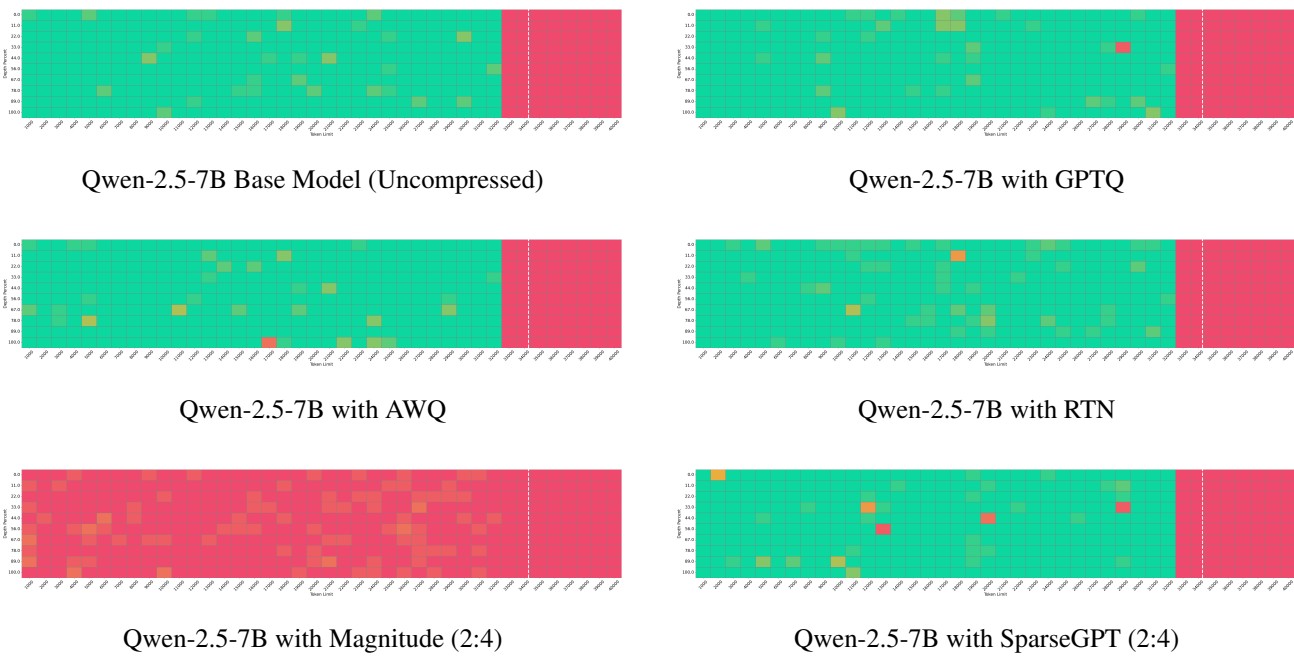

Qwen-2.5-7B Base Model (Uncompressed)

Qwen-2.5-7B with GPTQ

Qwen-2.5-7B with AWQ

Qwen-2.5-7B with RTN

Qwen-2.5-7B with Magnitude (2:4)

Qwen-2.5-7B with SparseGPT (2:4)

Figure 12: Comparative analysis of attention patterns in Qwen-2.5-7B under different compression schemes. The visualizations illustrate the preservation or degradation of attention mechanisms across a 40k token context window, highlighting the varying impacts of different compression methods on long-range dependency modeling.

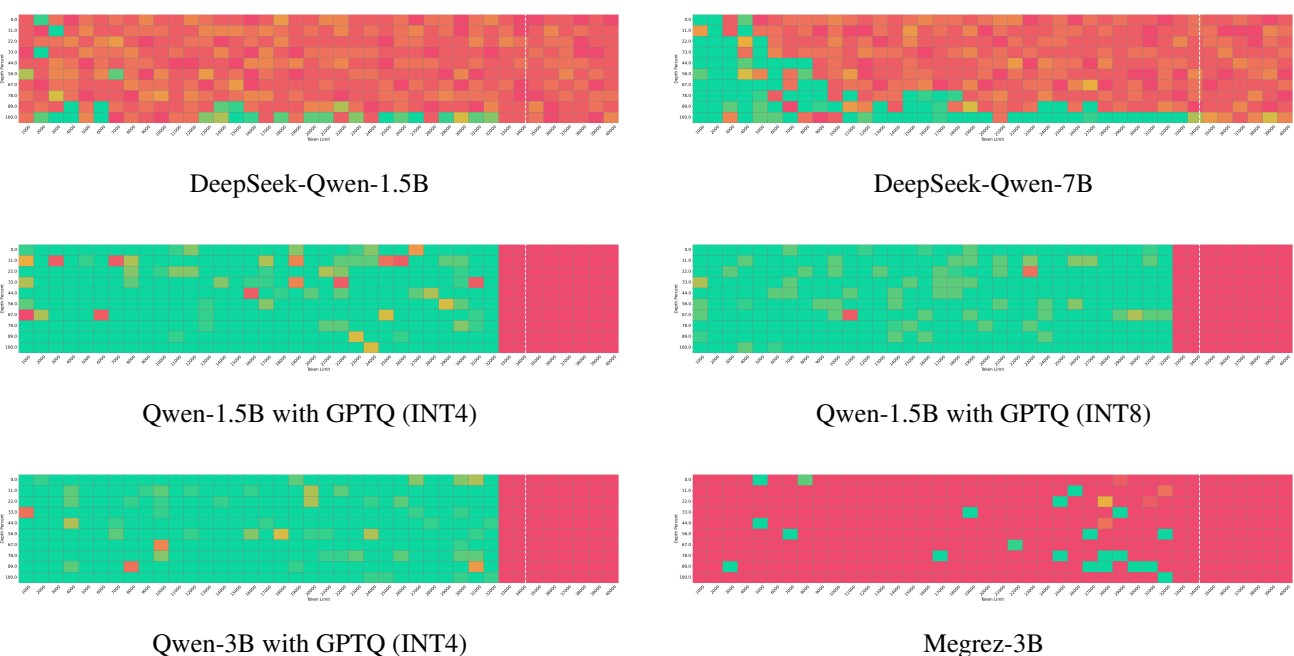

DeepSeek-Qwen-1.5B

DeepSeek-Qwen-7B

Qwen-1.5B with GPTQ (INT4)

Qwen-1.5B with GPTQ (INT8)

Qwen-3B with GPTQ (INT4)

Megrez-3B

Figure 13: Attention pattern analysis for smaller and distilled models across 40k context length. The visualizations demonstrate the varying capabilities of different model architectures and sizes in maintaining coherent attention patterns over long sequences, with particular focus on the effects of model distillation and parameter efficiency.

Table 8: Performance Comparison of Quantization and Sparsification Methods on Tool Use Tasks under String and JSON Formats

| LLMs | Compression | Instruct | | Plan | | Reason | | Retrieve | | Understand | | Review Choice | Overall |
|---|---|---|---|---|---|---|---|---|---|---|---|---|---|
| | | String | Json | String | Json | String | Json | String | Json | String | Json | | |
| InternLM2.5-7B | Mag(2:4) | 27.2 | 13.3 | 29.3 | 0.6 | 41.4 | 5.0 | 73.2 | 1.6 | 62.8 | 7.9 | 10.5 | 24.8 |
| | Mag(Un) | 57.8 | 73.2 | 27.7 | 23.1 | 59.2 | 19.6 | 86.4 | 20.6 | 73.2 | 24.2 | 60.6 | 47.8 |
| | SparseGPT(2:4) | 73.2 | 33.3 | 26.6 | 0.1 | 52.0 | 12.8 | 71.1 | 8.0 | 41.6 | 9.8 | 64.1 | 35.7 |
| | SparseGPT(Un) | 84.5 | 80.3 | 42.4 | 62.3 | 63.6 | 39.7 | 90.9 | 47.4 | 74.7 | 41.5 | 57.3 | 62.2 |
| | Wanda(2:4) | 78.5 | 46.4 | 50.6 | 11.2 | 59.6 | 21.9 | 95.8 | 18.1 | 59.7 | 21.4 | 61.4 | 47.7 |
| | Wanda(Un) | 83.7 | 90.6 | 49.0 | 72.4 | 66.3 | 32.6 | 96.5 | 42.2 | 76.4 | 39.5 | 62.0 | 64.7 |
| | AWQ | 98.6 | 98.7 | 48.5 | 45.3 | 65.8 | 46.5 | 85.6 | 66.7 | 79.6 | 56.3 | 63.2 | 68.6 |
| | FP16 | 98.6 | 98.6 | 44.3 | 73.7 | 67.5 | 51.0 | 85.7 | 66.4 | 81.9 | 56.0 | 70.4 | 72.2 |
| | FP8 | 98.6 | 98.6 | 44.3 | 73.7 | 65.9 | 50.3 | 83.3 | 65.5 | 79.4 | 55.1 | 70.4 | 71.4 |
| | GPTQ | 96.5 | 98.4 | 53.9 | 76.2 | 66.9 | 44.5 | 93.7 | 56.9 | 81.2 | 49.5 | 72.5 | 71.8 |
| | Base | 98.6 | 98.6 | 44.3 | 73.7 | 65.9 | 50.3 | 83.3 | 65.5 | 79.4 | 55.1 | 70.4 | 71.4 |
| Mistral-7B | Mag(2:4) | 3.9 | 0.3 | 22.5 | 0.7 | 38.1 | 0.5 | 45.4 | 0.2 | 0.0 | 1.0 | 16.0 | 11.7 |
| | Mag(Un) | 0.4 | 4.2 | 44.9 | 33.4 | 42.5 | 3.8 | 37.1 | 7.1 | 0.0 | 6.0 | 54.4 | 21.3 |
| | SparseGPT(2:4) | 3.6 | 15.3 | 43.6 | 9.9 | 46.1 | 3.5 | 41.0 | 2.8 | 0.1 | 3.7 | 10.9 | 16.4 |
| | SparseGPT(Un) | 92.0 | 40.8 | 59.2 | 56.6 | 45.8 | 8.2 | 28.9 | 13.9 | 0.0 | 13.7 | 77.8 | 39.7 |
| | Wanda(2:4) | 74.4 | 30.8 | 42.1 | 51.0 | 44.9 | 2.5 | 48.2 | 1.0 | 0.1 | 4.3 | 21.6 | 29.2 |
| | Wanda(Un) | 88.8 | 27.3 | 61.8 | 70.1 | 47.0 | 7.5 | 29.3 | 9.8 | 0.1 | 9.0 | 68.4 | 38.1 |
| | AWQ | 28.7 | 0.0 | 43.9 | 0.0 | 53.0 | 4.7 | 85.4 | 7.3 | 43.9 | 7.2 | 1.8 | 25.1 |
| | FP16 | 28.7 | 0.0 | 43.9 | 0.0 | 54.8 | 5.0 | 88.1 | 7.5 | 45.2 | 7.4 | 1.8 | 25.7 |
| | FP8 | 28.7 | 0.0 | 43.9 | 0.0 | 53.0 | 4.7 | 85.4 | 7.3 | 43.9 | 7.2 | 1.8 | 25.1 |
| | GPTQ | 28.7 | 0.0 | 43.9 | 0.0 | 53.0 | 4.7 | 85.4 | 7.3 | 43.9 | 7.2 | 1.8 | 25.1 |
| Qwen2.5-14B | FP16 | 98.8 | 98.5 | 68.6 | 82.4 | 65.0 | 64.3 | 94.5 | 78.7 | 69.9 | 67.4 | 64.3 | 77.5 |
| | Mag(2:4) | 11.7 | 2.8 | 29.3 | 4.0 | 32.6 | 5.3 | 62.5 | 5.7 | 34.8 | 4.5 | 17.0 | 19.1 |
| | Mag(Un) | 85.8 | 59.6 | 59.4 | 54.8 | 49.0 | 15.1 | 74.4 | 19.5 | 50.4 | 15.0 | 72.9 | 50.5 |
| | SparseGPT(2:4) | 80.0 | 43.9 | 52.1 | 7.0 | 60.4 | 31.1 | 73.0 | 24.2 | 68.3 | 27.0 | 83.0 | 50.0 |
| | SparseGPT(Un) | 98.3 | 98.5 | 67.4 | 84.8 | 63.1 | 60.3 | 89.4 | 77.2 | 67.6 | 67.3 | 73.5 | 77.0 |
| | Wanda(2:4) | 90.0 | 63.3 | 61.0 | 73.8 | 61.0 | 45.2 | 90.4 | 51.5 | 69.6 | 43.2 | 78.6 | 66.1 |
| Qwen2.5-7B | AWQ | 95.0 | 53.3 | 67.6 | 69.8 | 63.5 | 57.5 | 94.9 | 72.0 | 62.5 | 62.4 | 72.9 | 70.1 |
| | FP16 | 95.0 | 53.3 | 67.0 | 66.7 | 63.5 | 57.8 | 94.8 | 72.1 | 62.4 | 62.8 | 74.1 | 70.0 |
| | FP8 | 95.0 | 53.3 | 67.0 | 66.7 | 63.5 | 57.8 | 94.8 | 72.1 | 62.4 | 62.8 | 74.1 | 70.0 |
| | GPTQ | 80.5 | 75.7 | 65.3 | 72.8 | 62.1 | 59.3 | 91.4 | 74.7 | 63.9 | 64.8 | 70.4 | 71.0 |
| | Mag(2:4) | 0.0 | 0.0 | 7.0 | 0.0 | 14.8 | 0.0 | 20.0 | 0.0 | 0.9 | 0.1 | 0.6 | 3.9 |
| | Mag(Un) | 0.1 | 0.1 | 2.0 | 0.0 | 15.2 | 0.0 | 47.6 | 0.2 | 14.1 | 0.1 | 0.6 | 7.3 |
| | SparseGPT(2:4) | 27.5 | 23.4 | 59.2 | 6.1 | 55.8 | 33.2 | 90.2 | 34.5 | 68.4 | 32.0 | 60.0 | 44.6 |
| | SparseGPT(Un) | 56.3 | 52.6 | 66.2 | 68.2 | 61.3 | 50.0 | 92.5 | 55.6 | 65.1 | 48.1 | 67.8 | 62.2 |
| | Wanda(2:4) | 38.4 | 39.1 | 63.0 | 33.8 | 60.4 | 44.7 | 89.7 | 45.3 | 68.1 | 44.2 | 57.7 | 53.1 |
| | Wanda(Un) | 68.2 | 51.6 | 65.3 | 32.6 | 62.0 | 54.1 | 95.2 | 65.2 | 68.5 | 57.9 | 64.3 | 62.3 |
| DS-LLama-8B | FP16 | 6.1 | 1.0 | 37.7 | 15.2 | 60.3 | 34.3 | 76.8 | 31.7 | 0.0 | 28.0 | 8.8 | 27.3 |
| DS-Qwen-1.5B | FP16 | 10.8 | 13.8 | 33.3 | 8.7 | 42.4 | 24.7 | 55.2 | 16.9 | 49.6 | 16.6 | 5.1 | 25.2 |
| DS-Qwen-7B | FP16 | 44.6 | 54.0 | 42.2 | 22.8 | 54.1 | 42.3 | 71.6 | 37.8 | 66.7 | 34.0 | 9.2 | 43.6 |

Table 9: Task Performance Comparison Across Models and Compression Methods on AgentBoard

| LLMs | Compression | Jericho | | Pddl | | Tool-query | | Tool-Operation | | Scienceworld | |
|---|---|---|---|---|---|---|---|---|---|---|---|
| | | P | S | P | S | P | S | P | S | P | S |
| Qwen2.5-7B | - | 8.21% | 0.00% | 33.56% | 13.33% | 44.00% | 18.33% | 20.00% | 0.00% | 26.79% | 12.22% |
| | Wanda(Un) | 11.22% | 0.00% | 14.00% | 3.33% | 38.79% | 8.33% | 15.04% | 0.00% | 14.98% | 6.67% |
| | Wanda(2:4) | 1.25% | 0.00% | 0.56% | 0.00% | 11.86% | 0.00% | 3.79% | 0.00% | 1.80% | 0.00% |
| | SparseGPT(Un) | 17.39% | 0.00% | 20.13% | 3.33% | 35.87% | 8.33% | 16.58% | 0.00% | 14.78% | 3.33% |
| | SparseGPT(2:4) | 5.71% | 0.00% | 7.46% | 1.67% | 20.41% | 0.00% | 7.46% | 0.00% | 7.89% | 0.00% |
| | Mag(Un) | | | 0.00% | 0.00% | 0.00% | 0.00% | 0.00% | 0.00% | 0.00% | 0.00% |
| | Mag(2:4) | 0.00% | 0.00% | 0.00% | 0.00% | 0.00% | 0.00% | 0.00% | 0.00% | 0.00% | 0.00% |
| | GPTQ | 12.56% | 0.00% | 22.89% | 10.00% | 42.27% | 15.00% | 15.71% | 0.00% | 26.41% | 13.33% |
| | AWQ | 19.73% | 0.00% | 27.14% | 13.33% | 47.16% | 25.00% | 21.46% | 0.00% | 17.33% | 5.56% |
| InternLM2.5-7B | - | 14.23% | 0.00% | 17.64% | 0.00% | 31.64% | 8.33% | 7.67% | 0.00% | 13.91% | 2.22% |
| | Wanda(Un) | 6.96% | 0.00% | 9.96% | 1.67% | 30.67% | 3.33% | 18.00% | 0.00% | 9.79% | 1.11% |
| | Wanda(2:4) | 0.71% | 0.00% | 10.72% | 1.67% | 10.28% | 1.67% | 14.92% | 0.00% | 0.00% | 0.00% |
| | SparseGPT(Un) | 6.34% | 0.00% | 3.67% | 0.00% | 18.89% | 0.00% | 14.63% | 0.00% | 8.98% | 2.22% |
| | SparseGPT(2:4) | 0.00% | 0.00% | 0.00% | 0.00% | 15.77% | 0.00% | 1.25% | 0.00% | 0.00% | 0.00% |
| | Mag(Un) | 4.55% | 0.00% | 4.44% | 1.67% | 18.40% | 0.00% | 14.00% | 0.00% | 3.58% | 0.00% |
| | Mag(2:4) | 4.58% | 0.00% | 0.00% | 0.00% | 11.03% | 0.00% | 1.33% | 0.00% | 0.00% | 0.00% |
| | GPTQ | 12.68% | 0.00% | 25.29% | 6.67% | 21.69% | 5.00% | 14.21% | 0.00% | 15.12% | 4.44% |
| | AWQ | 11.22% | 5.00% | 18.67% | 3.33% | 34.48% | 8.33% | 2.08% | 0.00% | 13.81% | 3.33% |
| DS-Qwen-7b | Distilled | 1.96% | 0.00% | 1.11% | 0.00% | 8.52% | 0.00% | 2.67% | 0.00% | 2.57% | 0.00% |
| DS-Qwen-1.5b | Distilled | 3.21% | 0.00% | 0.00% | 0.00% | 5.65% | 0.00% | 4.67% | 0.00% | 0.00% | 0.00% |
| DS-LLama-8b | Distilled | 0.00% | 0.00% | 0.00% | 0.00% | 1.07% | 0.00% | 0.87% | 0.00% | 1.24% | 0.00% |
| Qwen2.5-3B | GPTQ | 11.31% | 0.00% | 14.61% | 3.33% | 30.37% | 11.67% | 13.20% | 0.00% | 17.40% | 3.20% |
| | AWQ | 8.84% | 0.00% | 23.35% | 6.67% | 30.65% | 6.67% | 7.00% | 0.00% | 16.82% | 2.22% |

Table 10: Task Performance Comparison Across Models and Compression Methods on WorfBench

| LLMs | Compression | Alfworld | | | Lumos | | | OS | | | ToolAlpaca | | | ToolBench | | | Webshop | | | Average | | |
|---|---|---|---|---|---|---|---|---|---|---|---|---|---|---|---|---|---|---|---|---|---|---|
| | | P | R | F1 | P | R | F1 | P | R | F1 | P | R | F1 | P | R | F1 | P | R | F1 | P | R | F1 |
| Qwen2.5-7B | SmoothQ(W8A8) | 0.46 | 0.56 | 0.48 | 0.51 | 0.64 | 0.57 | 0.77 | 0.92 | 0.81 | 0.03 | 0.04 | 0.03 | 0.00 | 0.00 | 0.00 | 0.81 | 0.70 | 0.75 | 0.43 | 0.48 | 0.44 |
| | GPTQ(INT4) | 0.48 | 0.54 | 0.48 | 0.50 | 0.63 | 0.56 | 0.77 | 0.93 | 0.83 | 0.02 | 0.02 | 0.02 | 0.00 | 0.00 | 0.00 | 0.81 | 0.69 | 0.74 | 0.43 | 0.47 | 0.44 |
| | AWQ(INT4) | 0.55 | 0.58 | 0.54 | 0.51 | 0.64 | 0.57 | 0.77 | 0.91 | 0.81 | 0.07 | 0.07 | 0.07 | 0.00 | 0.00 | 0.00 | 0.81 | 0.70 | 0.74 | 0.45 | 0.48 | 0.46 |
| | Wanda(Un) | 0.55 | 0.55 | 0.52 | 0.67 | 0.73 | 0.70 | 0.73 | 0.84 | 0.75 | 0.08 | 0.08 | 0.08 | 0.00 | 0.00 | 0.00 | 0.79 | 0.69 | 0.73 | 0.47 | 0.48 | 0.46 |
| | Wanda(2:4) | 0.53 | 0.49 | 0.49 | 0.63 | 0.68 | 0.65 | 0.82 | 0.84 | 0.79 | 0.09 | 0.09 | 0.08 | 0.00 | 0.00 | 0.00 | 0.78 | 0.67 | 0.71 | 0.47 | 0.46 | 0.46 |
| | SparseGPT(Un) | 0.49 | 0.51 | 0.48 | 0.62 | 0.68 | 0.65 | 0.78 | 0.91 | 0.82 | 0.08 | 0.07 | 0.07 | 0.00 | 0.00 | 0.00 | 0.80 | 0.71 | 0.74 | 0.46 | 0.48 | 0.46 |
| | SparseGPT(2:4) | 0.60 | 0.45 | 0.50 | 0.59 | 0.66 | 0.63 | 0.82 | 0.81 | 0.78 | 0.09 | 0.10 | 0.09 | 0.00 | 0.00 | 0.00 | 0.82 | 0.70 | 0.75 | 0.49 | 0.45 | 0.46 |
| Qwen2.5-3B | Mag(2:4) | 0.00 | 0.00 | 0.00 | 0.00 | 0.00 | 0.00 | 0.00 | 0.00 | 0.00 | 0.00 | 0.00 | 0.00 | 0.00 | 0.00 | 0.00 | 0.00 | 0.00 | 0.00 | 0.00 | 0.00 | 0.00 |
| | GPTQ(INT8) | 0.39 | 0.36 | 0.36 | 0.41 | 0.39 | 0.40 | 0.72 | 0.87 | 0.76 | 0.61 | 0.73 | 0.64 | 0.68 | 0.76 | 0.68 | 0.76 | 0.64 | 0.68 | 0.60 | 0.62 | 0.59 |
| | GPTQ(INT4) | 0.37 | 0.35 | 0.34 | 0.39 | 0.37 | 0.38 | 0.69 | 0.90 | 0.76 | 0.48 | 0.68 | 0.54 | 0.66 | 0.73 | 0.66 | 0.73 | 0.62 | 0.64 | 0.55 | 0.61 | 0.55 |
| Qwen2.5-32B | AWQ(INT4) | 0.41 | 0.42 | 0.40 | 0.42 | 0.38 | 0.40 | 0.75 | 0.86 | 0.78 | 0.57 | 0.71 | 0.61 | 0.69 | 0.76 | 0.69 | 0.28 | 0.30 | 0.28 | 0.52 | 0.57 | 0.53 |
| | GPTQ(INT4) | 0.52 | 0.58 | 0.53 | 0.64 | 0.64 | 0.64 | 0.81 | 0.89 | 0.83 | 0.69 | 0.68 | 0.66 | 0.78 | 0.77 | 0.75 | 0.80 | 0.69 | 0.74 | 0.71 | 0.71 | 0.69 |
| | AWQ(INT4) | 0.56 | 0.62 | 0.57 | 0.69 | 0.68 | 0.69 | 0.82 | 0.90 | 0.83 | 0.74 | 0.71 | 0.70 | 0.74 | 0.77 | 0.72 | 0.81 | 0.69 | 0.74 | 0.72 | 0.73 | 0.71 |
| Qwen2.5-14B | - | 0.55 | 0.62 | 0.57 | 0.69 | 0.68 | 0.68 | 0.84 | 0.91 | 0.85 | 0.74 | 0.71 | 0.70 | 0.76 | 0.79 | 0.75 | 0.82 | 0.71 | 0.75 | 0.73 | 0.74 | 0.72 |
| | GPTQ(INT4) | 0.28 | 0.22 | 0.24 | 0.36 | 0.38 | 0.37 | 0.36 | 0.54 | 0.42 | 0.43 | 0.40 | 0.40 | 0.24 | 0.30 | 0.26 | 0.49 | 0.42 | 0.45 | 0.36 | 0.38 | 0.36 |
| | AWQ(INT4) | 0.60 | 0.58 | 0.57 | 0.76 | 0.69 | 0.71 | 0.84 | 0.89 | 0.82 | 0.66 | 0.66 | 0.63 | 0.70 | 0.66 | 0.66 | 0.80 | 0.70 | 0.74 | 0.72 | 0.69 | 0.69 |
| Qwen2.5-1.5B | - | 0.55 | 0.59 | 0.54 | 0.71 | 0.70 | 0.70 | 0.82 | 0.88 | 0.83 | 0.63 | 0.67 | 0.62 | 0.70 | 0.74 | 0.73 | 0.79 | 0.67 | 0.72 | 0.71 | 0.71 | 0.69 |
| | GPTQ(INT8) | 0.62 | 0.52 | 0.54 | 0.66 | 0.71 | 0.68 | 0.80 | 0.84 | 0.79 | 0.44 | 0.66 | 0.50 | 0.44 | 0.68 | 0.51 | 0.81 | 0.70 | 0.75 | 0.63 | 0.68 | 0.63 |
| | GPTQ(INT4) | 0.63 | 0.45 | 0.51 | 0.70 | 0.69 | 0.69 | 0.77 | 0.84 | 0.78 | 0.45 | 0.64 | 0.50 | 0.50 | 0.73 | 0.58 | 0.82 | 0.71 | 0.76 | 0.65 | 0.68 | 0.64 |
| Mistral-7B-v0.3 | AWQ(INT4) | 0.61 | 0.50 | 0.53 | 0.61 | 0.69 | 0.65 | 0.78 | 0.82 | 0.77 | 0.39 | 0.69 | 0.48 | 0.39 | 0.62 | 0.46 | 0.82 | 0.70 | 0.75 | 0.60 | 0.67 | 0.61 |
| | SmoothQ(W8A8) | 0.64 | 0.63 | 0.61 | 0.66 | 0.68 | 0.67 | 0.67 | 0.85 | 0.72 | 0.00 | 0.00 | 0.00 | 0.00 | 0.00 | 0.00 | 0.77 | 0.70 | 0.73 | 0.46 | 0.48 | 0.46 |
| | GPTQ(INT8) | 0.62 | 0.63 | 0.61 | 0.67 | 0.70 | 0.68 | 0.66 | 0.83 | 0.71 | 0.00 | 0.00 | 0.00 | 0.00 | 0.00 | 0.00 | 0.77 | 0.70 | 0.72 | 0.45 | 0.48 | 0.45 |
| | GPTQ(INT4) | 0.54 | 0.57 | 0.54 | 0.66 | 0.69 | 0.68 | 0.67 | 0.86 | 0.73 | 0.00 | 0.00 | 0.00 | 0.00 | 0.00 | 0.00 | 0.80 | 0.70 | 0.74 | 0.45 | 0.47 | 0.45 |
| InternLM-2.5-7B | AWQ(INT4) | 0.59 | 0.63 | 0.59 | 0.67 | 0.70 | 0.69 | 0.72 | 0.86 | 0.76 | 0.00 | 0.00 | 0.00 | 0.00 | 0.00 | 0.00 | 0.79 | 0.69 | 0.73 | 0.46 | 0.48 | 0.46 |
| | Wanda(Un) | 0.00 | 0.00 | 0.00 | 0.00 | 0.00 | 0.00 | 0.69 | 0.75 | 0.69 | 0.00 | 0.00 | 0.00 | 0.00 | 0.00 | 0.00 | 0.00 | 0.00 | 0.00 | 0.11 | 0.13 | 0.12 |
| | Wanda(2:4) | 0.20 | 0.19 | 0.19 | 0.01 | 0.00 | 0.01 | 0.74 | 0.77 | 0.73 | 0.00 | 0.00 | 0.00 | 0.00 | 0.00 | 0.00 | 0.41 | 0.35 | 0.38 | 0.23 | 0.22 | 0.22 |
| | SparseGPT(Un) | 0.48 | 0.46 | 0.46 | 0.01 | 0.01 | 0.01 | 0.71 | 0.83 | 0.74 | 0.00 | 0.00 | 0.00 | 0.00 | 0.00 | 0.00 | 0.00 | 0.00 | 0.00 | 0.20 | 0.22 | 0.20 |
| | SparseGPT(2:4) | 0.33 | 0.47 | 0.38 | 0.34 | 0.36 | 0.34 | 0.63 | 0.78 | 0.67 | 0.00 | 0.00 | 0.00 | 0.00 | 0.00 | 0.00 | 0.76 | 0.65 | 0.70 | 0.34 | 0.38 | 0.35 |
| | Mag(Un) | 0.00 | 0.00 | 0.00 | 0.00 | 0.00 | 0.00 | 0.54 | 0.46 | 0.48 | 0.00 | 0.00 | 0.00 | 0.00 | 0.00 | 0.00 | 0.00 | 0.00 | 0.00 | 0.09 | 0.08 | 0.08 |
| | Mag(2:4) | 0.00 | 0.00 | 0.00 | 0.00 | 0.00 | 0.00 | 0.68 | 0.65 | 0.64 | 0.00 | 0.00 | 0.00 | 0.00 | 0.00 | 0.00 | 0.53 | 0.45 | 0.48 | 0.20 | 0.18 | 0.19 |
| DS-R1-Distill-Qwen-7B | Distilled | 0.00 | 0.00 | 0.00 | 0.01 | 0.01 | 0.01 | 0.69 | 0.88 | 0.75 | 0.12 | 0.22 | 0.15 | 0.24 | 0.40 | 0.28 | 0.00 | 0.00 | 0.00 | 0.18 | 0.25 | 0.20 |
| DS-R1-Distill-Qwen-1.5B | Distilled | 0.16 | 0.30 | 0.20 | 0.54 | 0.63 | 0.56 | 0.69 | 0.85 | 0.73 | 0.24 | 0.51 | 0.32 | 0.05 | 0.07 | 0.05 | 0.68 | 0.73 | 0.71 | 0.41 | 0.51 | 0.43 |
| DS-R1-Distill-Llama-8B | Distilled | 0.05 | 0.04 | 0.05 | 0.01 | 0.01 | 0.01 | 0.80 | 0.81 | 0.77 | 0.16 | 0.28 | 0.19 | 0.31 | 0.44 | 0.34 | 0.36 | 0.31 | 0.33 | 0.28 | 0.31 | 0.28 |
| Megrez-3Btruct | - | 0.00 | 0.00 | 0.00 | 0.00 | 0.00 | 0.00 | 0.70 | 0.82 | 0.73 | 0.00 | 0.00 | 0.00 | 0.00 | 0.00 | 0.00 | 0.00 | 0.00 | 0.00 | 0.12 | 0.14 | 0.12 |

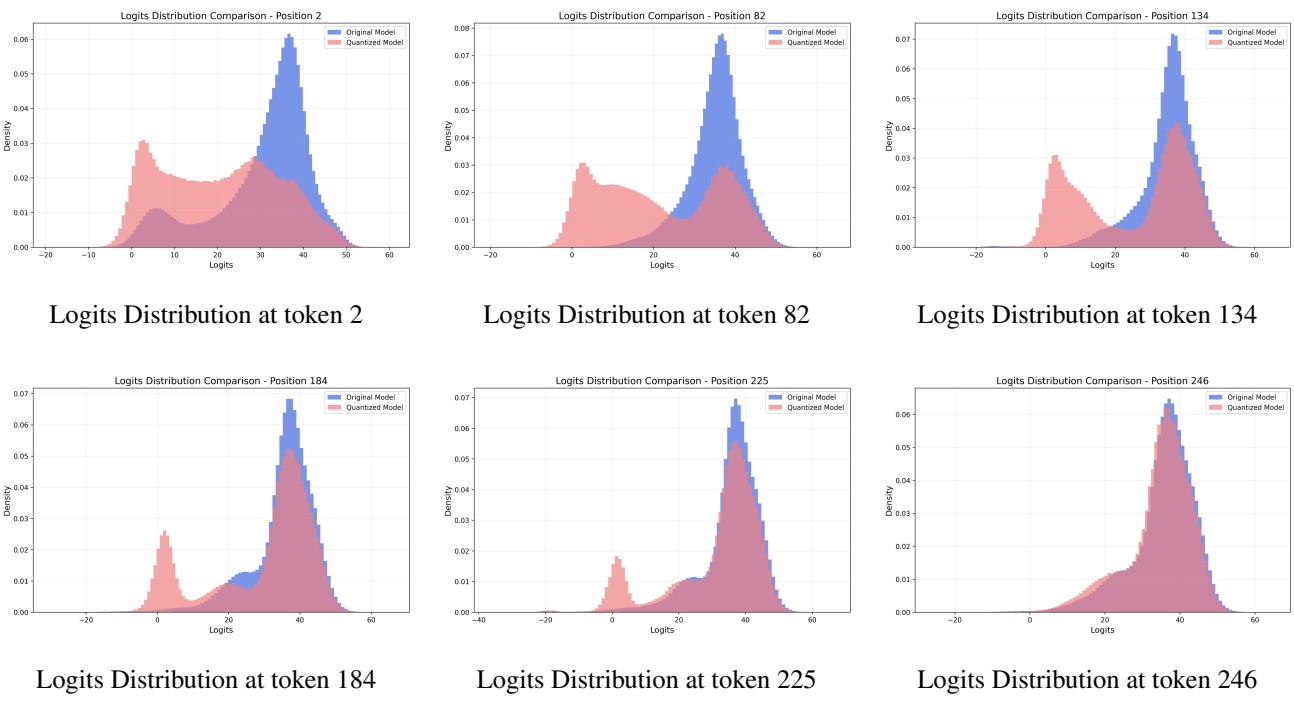

Logits Distribution at token 2 — Logits Distribution at token 82 — Logits Distribution at token 134

Logits Distribution at token 184 — Logits Distribution at token 225 — Logits Distribution at token 246

Figure 14: Logits distribution visualization across different sequence positions in compressed models

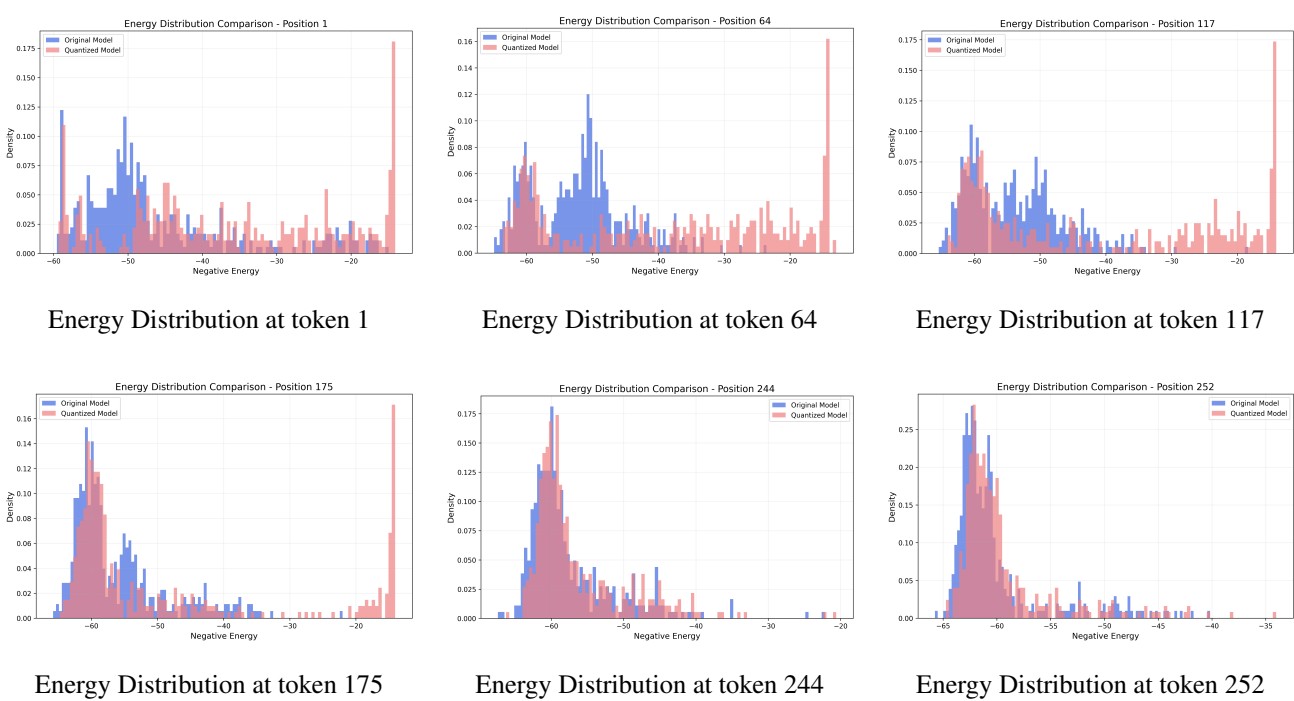

Energy Distribution at token 1 — Energy Distribution at token 64 — Energy Distribution at token 117

Energy Distribution at token 175 — Energy Distribution at token 244 — Energy Distribution at token 252

Figure 15: Energy distribution analysis at various sequence positions in compressed models

