# OpenReview forum: "Can Compressed LLMs Truly Act? An Empirical Evaluation of Agentic Capabilities in LLM Compression"
_ICML.cc/2025/Conference — ICML 2025 poster_

### Official Review · Reviewer_vD1Z · 2025-03-11

**Overall Recommendation:** 3

**Summary:**

The ACBench framework proposed is designed to systematically evaluate the effects of compression on both agent capabilities and large language models (LLMs). It tests agent capabilities across key areas such as action execution, workflow generation, long-context understanding, and real-world application performance. For LLMs, it evaluates efficient rank, top-K ranking consistency, and energy-based analysis to measure the impact on model efficiency and output reliability. Additionally, ACBench analyzes the impact of different compression approaches, providing insights into their trade-offs and suitability for various tasks. This framework serves as a robust tool for optimizing compressed models while maintaining performance and practicality.

**Claims And Evidence:**

Yes

**Essential References Not Discussed:**

1.	The compression methods are limited to only five compression methods, namely GPTQ, AWQ, SmoothQuant, SparseGPT, Wanda. (For Quantization and Pruning)
2.	Other compression methods like LoRA and Distillation are not systematically analyzed. (The passage use results from distilled model and original models  to evaluate the distillation methods but not thoroughly analyzed.)
3.	Larger Models (only contains size from 1.5B to 7B) are not evaluated.

**Experimental Designs Or Analyses:**

1.	It seems like the ACB mainly categorizes the existing Benchmark, while the one proposed by the essay, Action Execution aspect, lacks evaluation metrics setup.
2.	The evaluation in Workflow Generation and Real-World Application seem to have overshadowing parts about the Embodied AI task.

**Methods And Evaluation Criteria:**

YES

**Other Comments Or Suggestions:**

Please refer to the section Other Strengths and Weaknesses

**Other Strengths And Weaknesses:**

Strength
1.	The essay conducts thorough and comprehensive experiments and provide rather abundant elaboration on the methods and benchmarks it uses.
2.	The essay compares different models regarding different fields and uses multiple metrics to evaluate those agents’ ability.

Weakness
1.	Maybe need categorization for difference Benchmarks, like what capabilities are each Benchmark is testing. The results analyzed seem irrelevant (For example, you mention that four capabilities should be evaluated multi-step planning, long-context coherence, adaptive reasoning. How is each Benchmark related to these core aspects?)
2.	Some expressions are vague. For example, “Knowledge Distillation Shows Unexpected Performance Characteristics” is not a good summary sentence to conclude without showing what characteristic is.
3.	What is the relationship between the degradation of the compression has on LLMs and the degradation of the compression has on the Agentic Behaviors? You seem to analyze them separately, but more correlation analysis is expected.
4.	Optimal Compression Strategy Recommendations proposed in Chapter 4 should be analyzed after all the aspects are evaluated
5.	The framework stresses the importance of “Multi-Turn” conversation testing, but it is not highlighted in the following experiments

**Questions For Authors:**

Please refer to the section Other Strengths and Weaknesses

**Relation To Broader Scientific Literature:**

Integration of several Benchmarks and systematically propose the evaluation aspects for agentic behaviors.

**Theoretical Claims:**

Theoretical claims are poor for this article.

---

> ### Author Rebuttal · Authors · 2025-04-01
>
> **Q1**: About Theory.
>
> > Theoretical claims are poor for this article. What is the relationship between the degradation of the compression has on LLMs and the degradation of the compression has on the Agentic Behaviors?
> >
>
> ***Ans for Q1***: This paper is primarily an empirical study, as indicated by the title. To address this, we have developed a concise framework to explain how quantization errors propagate in sequential decision-making (agent scenarios). For details, please refer to this [anonymous link](https://anonymous.4open.science/r/ICML_ACBench_Rebuttal-B4DA/).
>
> **Q2**: About the setup for Action Execution
>
> > Action Execution aspect, lacks evaluation metrics setup.
> >
>
> ***Ans for Q2***: The Action Execution metrics (Sec. 4) adopt T-Eval's setting, evaluating **Plan** (similarity/LIS), **Reason/Understand** (Sentence-BERT), **Retrieve** (exact match), **Instruct** (format/parameter accuracy), and **Review** (classification accuracy).
>
> **Q3**: About the Embodied AI tasks.
>
> > The evaluation in Workflow Generation and Real-World Application seem to have overshadowing parts about the Embodied AI task.
> >
>
> ***Ans for Q3***:  In this paper, we employed **ScienceWorld** and **AlfWorld** as they inherently require structured, multi-step reasoning, making them ideal for evaluating *workflow generation*. BabyAI, while useful for basic instruction-following, lacks the complexity needed for higher-level planning assessment. Refer to Tab.8,9 and  [anonymous link](https://anonymous.4open.science/r/ICML_ACBench_Rebuttal-B4DA/) for more details.
>
> **Q4**: About the compression methods
>
> > The compression methods are limited to only five compression methods. LoRA and Distillation are not systematically analyzed.
> >
> > The passage use results from distilled model and original models to evaluate the distillation methods but not thoroughly analyzed.
> >
> >  Larger Models (only contains size from 1.5B to 7B) are not evaluated.
>
> ***Ans for Q4***: As presented in Sec.2.2 and Sec.8, we prioritize these compression methods for the following reasons:
>
> 1. **Practical Impact & Compatibility**: The selected methods (GPTQ, AWQ, SmoothQuant, SparseGPT, Wanda) are foundational and widely adopted in serving systems, which are supported by vLLM[3] and SGLang[4]. They are critical for real-world high-throughput serving (e.g., 10–24× speedups over HuggingFace).
> 2. **Distillation/LoRA**:  While we include distilled models (e.g., R1-Distill series) in baseline comparisons, a systematic evaluation of distillation or LoRA lies beyond our focus on post-training compression. These techniques inherently require retraining or architectural modifications.
> 3. **Model Scale**: We evaluate up to **Qwen2.5-32B** (Tab.9), but larger models (e.g., 70B) are prohibitively expensive (>1 month/run on 8 GPUs). Memory constraints in long-context agent tasks further limit scaling.
>
> **Q5**: About the evaluation
>
> > Four capabilities should be evaluated multi-step planning, long-context coherence, adaptive reasoning. How is each Benchmark related to these core aspects?
> >
>
> ***Ans for Q5***: Thank you for raising this important point. We explicitly connect each benchmark to the core capabilities in the following ways:
>
> - **Multi-step planning** is evaluated in **Workflow (Sec. 5)** and **Real-World Applications (Sec. 7)**.
> - **Long-context coherence** is assessed in **Long-Context (Sec. 6)**.
> - **Adaptive reasoning** is demonstrated in both **Workflow (Sec. 5)** and **Real-World Applications (Sec. 7)**.
>
> In line 35-52, we have already refined these tasks. We will further enhance clarity in the revised manuscript
>
> **Q6**: About the writing.
>
> > Some expressions are vague. For example, “Knowledge Distillation Shows Unexpected Performance Characteristics” is not a good summary sentence to conclude without showing what characteristic is.
> >
>
> ***Ans for Q6***:   To address this, we have revised the statement to:
>
> > *"Knowledge distillation from a reasoning model lead to performance degradation in agent scenarios."*
>
>
>
> **Q7**: About the Optimal Strategy:
>
> > Optimal Compression Strategy Recommendations proposed in Chapter 4 should be analyzed after all the aspects are evaluated
> >
>
> ***Ans for Q7***: To align with the feedback, I propose refining the section title to **"Compression Strategy Recommendations for Action Execution"**. And for overall guidings, please refer to **Q3 to Reviewer DuJV**.
>
>
>
> **Q8**: About the "Multi-Turn".
>
> > The framework stresses the importance of “Multi-Turn” conversation testing, but it is not highlighted in the following experiments
> >
>
> ***Ans for Q8***: Thank you for your feedback. The "Multi-Turn" is implicitly integrated into the experimental design:
>
> - In Sec. 5): Tasks in WorfBench inherently involve multi-turn interactions, as agents must dynamically plan and execute
> - In Sec.7, Benchmarks like Jericho and PDDL explicitly test multi-turn reasoning. For instance, agents must navigate 10+ conversational turns to solve complex puzzles

---

> > ### Comment · Reviewer_vD1Z · 2025-04-08
> >
> > Thank you for your detailed response, your reply has answered my doubts to some extent. I have raised my score.

---

> > > ### Author Response · Authors · 2025-04-08
> > >
> > > Dear Reviewer vD1Z,
> > >
> > > Thank you for your thoughtful feedback and for revising the scores following our rebuttal. We sincerely appreciate the time and effort you dedicated to evaluating our work. Your suggestions and response mean a great deal to us.
> > >
> > > Best regards,
> > >
> > > Authors of #6400

---

### Official Review · Reviewer_Sovg · 2025-03-11

**Overall Recommendation:** 5

**Summary:**

This is a very interesting paper that studies agentic capabilities in LLM compression. The authors have carefully selected a series of evaluation benchmarks that cover practical scenarios of agent manipulation to assess the performance drop after compression.

**Claims And Evidence:**

Yes, the claims are well supported by clear and convincing evidence.

**Essential References Not Discussed:**

No.

**Experimental Designs Or Analyses:**

The experimental design is extensive and well-structured, providing a detailed evaluation of all aspects of agentic abilities. It also tests across a variety of language models.

**Methods And Evaluation Criteria:**

The proposed evaluation criteria are entirely reasonable.

**Other Comments Or Suggestions:**

No.

**Other Strengths And Weaknesses:**

This paper is excellent. Well done!

**Questions For Authors:**

No.

**Relation To Broader Scientific Literature:**

This paper provides a valuable window for readers to understand the influence of compression on agentic workflows and serves as a guide on which compression methods to choose. It contains many valuable insights.

**Theoretical Claims:**

There is no theoretical claim.

---

> ### Author Rebuttal · Authors · 2025-04-01
>
> Dear Reviewer Sovg,
>
> Thank you for your thorough and constructive review of our work. We sincerely appreciate your recognition of the experimental rigor and practical relevance of our evaluation benchmarks, as well as your encouraging feedback. Your insights strongly support our goal of providing actionable guidance on the trade-offs in LLM compression for agentic workflows.
>
> We are pleased that the detailed results and visualizations in the appendix proved useful—ensuring methodological transparency was a key priority for us. Should you have any further suggestions or require additional clarifications for the final version, we would be happy to incorporate them.
>
> Thank you once again for your time and thoughtful evaluation. We look forward to any additional comments you may have.
>
> Best regards,
>
> Authors of #6400

---

### Official Review · Reviewer_DuJV · 2025-03-14

**Overall Recommendation:** 3

**Summary:**

Large language models (LLMs) have significantly advanced areas such as code synthesis and multi-agent collaboration; however, their practical deployment remains constrained due to substantial computational and memory requirements. Compression techniques, including pruning and quantization, effectively reduce model size but frequently neglect crucial agent capabilities like planning, coherence, and tool integration.

This paper proposes the Agent Compression Benchmark (ACB), aiming to comprehensively assess the effects of compression methods—pruning (SparseGPT, Wanda) and quantization (GPTQ, AWQ)—on LLMs. The benchmark specifically evaluates:

1. Agent Capabilities: Action execution, long-context coherence, and tool integration.

2. Model Impact: Assessed through ERank, Top-K Ranking Correlation, and energy-based analyses.

3. Compression Comparison: Evaluates a range of models (from <7B up to 32B parameters) to provide insights for selecting optimal compression strategies without significantly compromising agent performance.

**Claims And Evidence:**

The primary goal of this paper—to extensively evaluate compression impacts across multiple dimensions—is clearly presented, and the paper thoroughly quantifies these effects across various models and settings. However, the claim that the proposed metrics (ERank, Top-K Ranking Correlation, and energy-based methods) significantly enhance understanding of compression impacts is less convincing, as their practical utility remains unclear.

**Essential References Not Discussed:**

None that I am aware of.

**Experimental Designs Or Analyses:**

As noted in the methods section, the experiments are comprehensive and rigorous. However, the connection between the experimental outcomes and the practical implications of the proposed metrics requires additional clarification.

**Methods And Evaluation Criteria:**

The selected models, datasets, and metrics are comprehensive and well-chosen. Nevertheless, the paper does not clearly articulate how the proposed metrics meaningfully contribute to a broad quantitative analysis. More explanation is needed on how these metrics translate into practical insights across diverse workloads.

**Other Comments Or Suggestions:**

I find it challenging to clearly understand how the proposed metrics (ERank, Top-K Ranking Correlation, and energy-based metrics) correlate with the specific benchmarks' performance metrics. Certain compression techniques perform well in specific scenarios but underperform in others. Although trends in the proposed metrics are observable, it is unclear how practitioners should interpret these results practically to predict or understand performance across diverse workloads.

**Other Strengths And Weaknesses:**

See "Other Comments or Suggestions" below.

**Questions For Authors:**

Could you elaborate on the significance of your proposed metrics to the main contributions of the paper? Specifically, why are these metrics important, and how can practitioners leverage them effectively to make informed decisions when choosing appropriate compression techniques for novel tasks?

**Relation To Broader Scientific Literature:**

The significant contribution of this paper lies in presenting a curated benchmark suite for comprehensively evaluating different facets of model compression. However, the paper lacks deeper insights and guiding principles on how practitioners might effectively leverage these extensive observations to make informed decisions.

**Theoretical Claims:**

The theoretical justifications and statistical analyses provided by the authors are reasonable and supported effectively by motivational diagrams.

---

> ### Author Rebuttal · Authors · 2025-04-01
>
> **Q1**: About the practical utility of the proposed metrics:
>
> > the proposed metrics significantly enhance understanding of compression impacts is less convincing, as their practical utility remains unclear.
>
> **Ans for Q1**: . We would like to address each metric:
>
> - **ERank**: Diff-eRank[1] is a theoretically grounded method based on information theory and geometric principles. It analyzed differences between base and trained models using ERank. We extend it to quantized LLMs shows ERank's effectiveness with experimental results (extending Tab.1 of Diff-eRank), where higher ERank values correlate with better model performance (see below table).
> - **Top-K Ranking Correlation**: As detailed in Q1 to Reviewer K5g3, this metric provides meaningful insights by focusing on the model's behavior regarding top-k token ranking. The correlation analysis captures important aspects of the model's predictive distribution.
> - **Energy-based Analysis**: We find that compressed models' energy distributions gradually align with uncompressed ones over timesteps (Fig.14), showing that while quantization disrupts LLM representations, parameter redundancy compensates for the loss. Aggregated energy reflects this recovery and correlates with performance (see below table).
>
> | OPT          | 125M   | 1.3B   | 2.7B   | 6.7B   |
> | ------------ | ------ | ------ | ------ | ------ |
> | ACC          | 0.276  | 0.332  | 0.370  | 0.360  |
> | delta Loss   | 5.734  | 6.138  | 6.204  | 6.258  |
> | Diff-ERank   | 1.410  | 2.140  | 2.338  | 2.280  |
> | ERank (4bit) | 15.462 | 15.589 | 13.898 | 17.877 |
> | Energy       | 2.738  | 2.746  | 2.631  | 2.883  |
>
> [1] Lai Wei etc, Diff-eRank: A Novel Rank-Based Metric for Evaluating Large Language Models, NeurIPS 2024
>
>
> **Q2**: About the connection between the metrics with traditional metric.
>
> > the connection between the experimental outcomes and the practical implications of the proposed metrics requires additional clarification. How the proposed metrics correlate with the specific benchmarks' performance metrics.
>
> ***Ans for Q2***: Our proposed three metrics serve as complementary tools to traditional benchmarks, offering causal insights into performance variations. Specifically, they help explain why certain models achieve better or worse results on standard metrics like perplexity (PPL) or accuracy. Also, please refer to our response to **Reviewer K5g3 (Q1)**, where we provide additional experiments demonstrating how **Top-K Ranking Correlation** reflects changes in traditional benchmark performance. You can  check the [anonymous link](https://anonymous.4open.science/r/ICML_ACBench_Rebuttal-B4DA/) about the topk-ranking results.
>
> **Q3**: **Guiding Principles for Practitioners**
>
> > The paper lacks deeper insights and guiding principles on how practitioners might effectively leverage these extensive observations to make informed decisions.
>
> ***Ans for Q3***: We have refined our guidelines to provide clearer, actionable insights for practitioners:
>
> 1. For Specific Agent Capabilities:
>    - If targeting single capability like tool use, directly consult the task-specific results in Sections 4-7 to select the optimal compression method.
>    - eg: For tool use, AWQ preserves JSON-structured outputs better than GPTQ (Fig. 3); for long-context, AWQ surpass GPTQ in most of cases.
>
> 2. For General-Purpose Agent Deployment:
>    - Model Choice > Compression Method: Base model capability is critical. For instance, Qwen2.5 outperforms R1-Distill-Qwen2.5  in four capabilies (Tab. 8). Prioritize high-quality base models first.
>    - Default to AWQ:  AWQ shows stable performance across all benchmarks (workflow, tool use, long-context).
>    - If Quantization is Infeasible: Use Wanda (outperforms SparseGPT in 80% of cases).
>    - Avoid R1-Distill for Agents: Despite its reasoning strengths, it fails in real-world agent tasks (Fig. 7). Use quantized base models instead.
>
> 3. For hybrid scenarios (e.g., workflow + tool use):  Start with Qwen2.5-7B (strong base) → Apply AWQ.
>
> You can refer to [anonymous link](https://anonymous.4open.science/r/ICML_ACBench_Rebuttal-B4DA/) for guideline flow chart.
>
> **Q4**: About the importance：
>
> > Could you elaborate on the significance of your proposed metrics to the main contributions of the paper?
>
> ***Ans for Q4***:
>
> These metrics are foundational to our three key contributions:
>
> - ERank can reflect how the information are compressed from the perspective of information theory. **See Q1 to Reviewer DuJV**.
> - Topk Ranking Correlation is intuitive over how the compression influence sampling process. we also added additional experiments to show that topk ranking correlation have the ability to predict the downstream performance.  **See Q1 to Reviewer K5g3**.
> - Energy is helpful for us to understand what is going on during the decoding stages, where as the time goes, the distribution shifted to uncompressed one and become stable. **See Q1 to Reviewer DuJV**

---

> > ### Comment · Reviewer_DuJV · 2025-04-02
> >
> > Thank you for the explanation and the additional experiment. I do not have further questions, but given my lack of expertise on the specific compression topics, I will keep the weak accept rating and leave further judgements to AC and other reviewers.

---

> > > ### Author Response · Authors · 2025-04-08
> > >
> > > Dear Reviewer DuJV,
> > >
> > > We sincerely appreciate the time and effort you have dedicated to reviewing our manuscript. Your valuable insights and suggestions are greatly appreciated and will certainly help us improve our paper.
> > >
> > > Best regards,
> > >
> > > Authors of #6400

---

### Official Review · Reviewer_K5g3 · 2025-03-14

**Overall Recommendation:** 4

**Summary:**

The authors introduce ACBench (Agent Compression Benchmark), a benchmark designed to evaluate how compression techniques (quantization and pruning) affect the agentic capabilities of large language models (LLMs), such as multi-step planning, workflow generation, tool use, and long-context understanding. They assess compression across three dimensions: the effect on agentic capabilities, internal model changes (using ERank, Top-K Ranking Correlation, and Energy-based Analysis), and the comparative effectiveness of compression methods (quantization vs. pruning). The authors argue that traditional benchmarks neglect real-world, multi-turn scenarios, making ACBench relevant for practical deployment. Their findings show that quantization preserves structured tasks well but significantly degrades performance on complex real-world applications, whereas pruning typically performs worse. Finally, they also found that distilled reasoning models performed poorly on agentic tasks.

## update after rebuttal
Thank you for the rebuttal. I have upped my score.

**Claims And Evidence:**

The paper’s core claim—that current benchmarks inadequately capture the performance impacts of compression on agentic capabilities—is convincingly supported by extensive experiments. However, the surprising underperformance of distilled models lacks deeper investigation or explanation, leaving uncertainty about whether this degradation arises specifically from compression or reflects a broader difficulty models have with multi-step tasks.

**Essential References Not Discussed:**

NA

**Experimental Designs Or Analyses:**

The experimental design is comprehensive and sound, evaluating a broad range of compressed models and techniques across diverse agentic tasks. However, a key limitation is that the experiments do not explicitly separate baseline model limitations from compression-specific degradation, nor do they compare or correlate performance systematically with simpler, traditional benchmarks.

**Methods And Evaluation Criteria:**

The proposed benchmark (ACBench) and evaluation criteria clearly address the identified gap by focusing explicitly on agentic capabilities—such as multi-step planning and long-context understanding—which current benchmarks neglect. The use of novel metrics (ERank, Top-K Ranking Correlation, Energy-based Analysis) to analyze internal model changes is well justified, although these metrics are not explicitly validated against established measures. It would be valuable to directly compare these novel metrics with traditional benchmarks and simpler metrics (such as perplexity, single-turn accuracy, or standard ranking correlations) to confirm whether they provide distinct additional insights. Additionally, the lack of explicit correlation analysis with simpler, existing benchmarks limits the ability to judge ACBench’s unique value, leaving open questions about whether observed performance differences arise specifically from compression or reflect more general model limitations on multi-step tasks.

**Other Comments Or Suggestions:**

Figure 5 is hard to read, and Figure 6 seems incomplete?

**Other Strengths And Weaknesses:**

I like the general idea of the paper, but as expressed above, I am concerned about: 1) the lack of explicit validation and comparison of the novel metrics (ERank, Top-K Ranking Correlation, Energy-based Analysis) against simpler, existing benchmarks and metrics, leaving their unique added value unclear; and 2) the insufficient exploration and explanation of why distilled reasoning models underperform specifically on agentic tasks, raising ambiguity about whether this is due to compression techniques or more fundamental limitations of models in multi-step reasoning scenarios.

**Questions For Authors:**

1) Can you clarify if (and how) the novel metrics (ERank, Top-K Ranking Correlation, Energy-based Analysis) were validated against simpler, traditional benchmarks and metrics (e.g., perplexity, accuracy, simpler correlation metrics)?

2) Can you elaborate on why distilled reasoning models unexpectedly underperform specifically on agentic tasks? Is this issue related directly to compression methods, or is it indicative of more general difficulties these models have with multi-step reasoning?

**Relation To Broader Scientific Literature:**

The key contribution of this paper—introducing the ACBench to systematically evaluate compression effects on agentic capabilities—addresses a specific gap in the broader literature. Prior research extensively studied compression techniques (quantization, pruning, distillation) mainly on traditional NLP benchmarks like GLUE or perplexity-based tasks. However, this literature rarely investigates how these techniques impact more complex, interactive, and multi-step tasks. ACBench builds upon recent work on evaluating and benchmarking LLMs’ agentic capabilities (such as workflow generation, long-context comprehension, and tool use), explicitly linking the compression literature to the emerging field of interactive agents. By doing so, the paper connects two previously distinct streams of research—model compression efficiency and agentic AI performance—offering insights particularly relevant for practical LLM deployments in resource-constrained, real-world scenarios.

**Theoretical Claims:**

NA

---

> ### Author Rebuttal · Authors · 2025-04-01
>
> Thank you for the thorough comments and recognition of our work. We appreciate that you acknowledge our experiments are “comprehensive and sound”. Please see our responses to your questions and concerns below.
>
> **Q1**: About the metrics.
>
> > Can you clarify if (and how) the novel metrics (ERank, Top-K Ranking Correlation, Energy-based Analysis) were validated against simpler, traditional benchmarks and metrics (e.g., perplexity, accuracy, simpler correlation metrics)?
> >
>
> ***Ans for Q1***:
>
> As explained in the **Abstract and Introduction (lines 97-108)**, our proposed metrics **are not against** traditional evaluation metrics. Instead, they focus on compression explainability by revealing how compression influences LLM behavior and causes degradation. While traditional metrics only reflect the overall performance decay after compression, they cannot show **how compression specifically affects the model behavior.** For example, our Topk Ranking Consistency metric focuses on the inference feature of language models: when quantization/pruning causes he token ranking dislocation. For instance, in an instruction fine-tuning scenario, quantization can reverse the probability order of "of course" and "sorry" tokens, directly changing the output affective tendency.
>
> Besides, inspired by this question, we explored that **whether our metrics can be used not only to exploiting of compression, but also how to choose compression methods.**  We first conducted experiments on InternLM2.5-20B:
>
> | InternLM2.5 20B | PPL | Hotpot QA | TriviaQA | MultiNews | Lcc | SciWorld | Topk R |
> | --- | --- | --- | --- | --- | --- | --- | --- |
> | AWQ | 7.61 | 36.52 | 84.69 | 25.82 | 61.68 | 13.81 | 87.29 |
> | GPTQ | 7.59 | 18.53 | 56.04 | 24.21 | 56.62 | 15.21 | 84.15 |
> | Mag(Un) | 10.39 | 31.59 | 79.26 | 26.2 | 49.72 | 3.58 | 49.44 |
> | SparseGPT(Un) | 7.65 | 34.21 | 84.43 | 26.11 | 49.42 | 8.98 | 57.36 |
> | Wanda(Un) | 7.87 | 32.65 | 87.44 | 25.87 | 58.49 | 9.79 | 58.98 |
>
> Then, we compute the **correlation of traditional metric (acc, ppl) with topk ranking**, and find that the ranking consistency between ppl and topk ranking are relative high, meaning that topk ranking can reflect the evaluation performance in some extend.
>
> | Pearson | PPL | HotpotQA | TriviaQA | MultiNews | Lcc | SciWorld | Topk R |
> | --- | --- | --- | --- | --- | --- | --- | --- |
> | PPL | 1 | 0.098 | 0.088 | 0.417 | -0.551 | -0.848 | -0.636 |
> | Topk R | -0.636 | -0.323 | -0.457 | -0.664 | 0.756 | 0.928 | 1 |
>
> Compared with ppl, our topk ranking achieved generally better performance, indicating stronger predictive capability for downstream tasks.
>
> **Q2: About deeper insight of the unexpected degradation**
>
> > Can you elaborate on why distilled reasoning models unexpectedly underperform specifically on agentic tasks? Is this issue related directly to compression methods, or is it indicative of more general difficulties these models have with multi-step reasoning?
> >
>
> ***Ans for Q2***:  The underperformance stems from two key factors:
>
> - **Knowledge Gap in R1:** Prior Deepseek V3/R1 lacked agentic capabilities (like tool use) until the recent V3 0324 version. Therefore, the distillation process couldn’t transfer the agentic knowledge from Teacher R1 to student Qwen. This distillation process mainly focus on reasoning and dialogue skills (evidenced by Qwen2.5 32B's improvement from 50.0%to 72.6% in AIME2024)[1].
> - **Capacity Tradeoff in Distillation**: The distilled models tested have limited capacity. When distillation priorizes core reasoning skills (math), agentic skills (function call) may be deprioritized. From the perspective of information bottleneck perspective, we formulate it as:
>
>     $$
>     \mathcal{L} _ {\text{distill}} =I(\theta_S; y_{\text{reason}})-\beta I(\theta_S; y_{\text{agent}})+\lambda\|\theta_S\|
>     $$
>
>     where $I(\theta_S; y_{reason})$ maximizes reasoning performance, $\beta I(\theta_S; y_{\text{agent}})$ represents the penalty on agentic skill retention, $\lambda\|\theta_S\|$ enforces model compactness. So, for capacity-constrained LLMs, maximizing reasoning performance often suppresses agentic capabilities due to competition for parameter space. This suggests that larger models can mitigate this trade-off. In the Wanda (2:4) benchmark, Qwen2.5-14B experienced an 11.4% performance degradation, while Qwen2.5-7B saw a more significant drop of 17%(from Table 7).
>
>
> [1] DeepSeek AI, DeepSeek-R1: Incentivizing Reasoning Capability in LLMs via Reinforcement Learning
>
> **Q3:** About Figures
>
> > Figure 5 is hard to read, and Figure 6 seems incomplete?
> >
>
> ***Ans for Q3***: We apologize for the clarity issues. For Fig.5, we improved readability by using darker color. For Fig. 6, we removed the redundant metric and add missing one(SmoothQ) to ensure completeness. You can  check the [anonymous link](https://anonymous.4open.science/r/ICML_ACBench_Rebuttal-B4DA/) over the updated figures.

---

> > ### Comment · Reviewer_K5g3 · 2025-04-02
> >
> > Thank you for the detailed responses and the clarification in Q2. The additional results wrt Q1 are especially interesting. I have updated my score.

---

> > > ### Author Response · Authors · 2025-04-03
> > >
> > > Dear reviewer K5g3,
> > >
> > > Thank you for your feedback and for raising scores following the rebuttal. We truly appreciate your time and consideration. Your continued support mean a great deal to us. Your insights have been instrumental in refining our work, thanks again for inspiring us (in Q1).
> > >
> > > Best regards,
> > > Authors of #6400

---

### Decision · Program_Chairs · 2025-05-01

**Decision:**

Accept (poster)

**Comment:**

This paper proposes a new evaluation framework called ACBench. It assesses how post-training compression methods affect the agentic capabilities of large language models. The advantages include: (1) It addresses a clear gap in evaluating agent-like behaviors in LLMs post-compression; (2) it provides actionable insights for deploying compressed LLMs in real-world multi-turn scenarios; (3) it compares various compression techniques on small to medium models. However, the major concern is that this paper is empirical and lacks theoretical formalism to explain observed effects, especially compression vs agentic degradation. Another concern is that it focuses mostly on <32B models and omits other methods like LoRA or larger models. Their practical utility remains partially unclear to some reviewers. Overall, this is a good paper and should be considered if there is room in the program.